# Learning Generalizable Visual Representations via Interactive Gameplay

**Luca Weihs**[1], **Aniruddha Kembhavi**[1], **Kiana Ehsani**[1], **Sarah M Pratt**[2], **Winson Han**[1], **Alvaro Herrasti**[1], **Eric Kolve**[1], **Dustin Schwenk**[1], **Roozbeh Mottaghi**[1], **Ali Farhadi**[2]

[1]Allen Institute for Artificial Intelligence
[2]University of Washington
[1]{lucaw,anik,kianae,winsonh,alvaroh,erick,dustins,roozbehm}@allenai.org
[2]{spratt3,ali}@cs.washington.edu

## Abstract

A growing body of research suggests that embodied gameplay, prevalent not just in human cultures but across a variety of animal species including turtles and ravens, is critical in developing the neural flexibility for creative problem solving, decision making, and socialization. Comparatively little is known regarding the impact of embodied gameplay upon artificial agents. While recent work has produced agents proficient in abstract games, these environments are far removed from the real world and thus these agents can provide little insight into the advantages of embodied play. Hiding games, such as hide-and-seek, played universally, provide a rich ground for studying the impact of embodied gameplay on representation learning in the context of perspective taking, secret keeping, and false belief understanding. Here we are the first to show that embodied adversarial reinforcement learning agents playing Cache, a variant of hide-and-seek, in a high fidelity, interactive, environment, learn generalizable representations of their observations encoding information such as object permanence, free space, and containment. Moving closer to biologically motivated learning strategies, our agents' representations, enhanced by intentionality and memory, are developed through interaction and play. These results serve as a model for studying how facets of vision develop through interaction, provide an experimental framework for assessing what is learned by artificial agents, and demonstrates the value of moving from large, static, datasets towards experiential, interactive, representation learning.

## 1 Introduction

We are interested in studying what facets of their environment artificial agents learn to represent through interaction and gameplay. We study this question within the context of hide-and-seek, for which proficiency requires an ability to navigate around in an environment and manipulate objects as well as an understanding of visual relationships, object affordances, and perspective. Inspired by behavior observed in juvenile ravens (Burghardt, 2005), we focus on a variant of hide-and-seek called *cache* in which agents hide objects instead of themselves. Advances in deep reinforcement learning have shown that, in abstract games (*e.g.* Go and Chess) and visually simplistic environments (*e.g.* Atari and grid-worlds) with limited interaction, artificial agents exhibit surprising emergent behaviours that enable proficient gameplay (Mnih et al., 2015; Silver et al., 2017); indeed, recent work (Chen et al., 2019; Baker et al., 2020) has shown this in the context of hiding games. Our interest, however, is in understanding how agents learn to represent their visual environment, through gameplay that requires varied interaction, in a high-fidelity environment grounded in the real world. This requires a fundamental shift away from existing popular environments and a rethinking of how the capabilities of artificial agents are evaluated.

Our agents must first be embodied within an environment allowing for diverse interaction and providing rich visual output. For this we leverage AI2-THOR (Kolve et al., 2017), a near photo-realistic, interactive, simulated, 3D environment of indoor living spaces, see Fig. 1a. Our agents are parameterized using deep neural networks, and trained adversarially using the paradigms of reinforce-

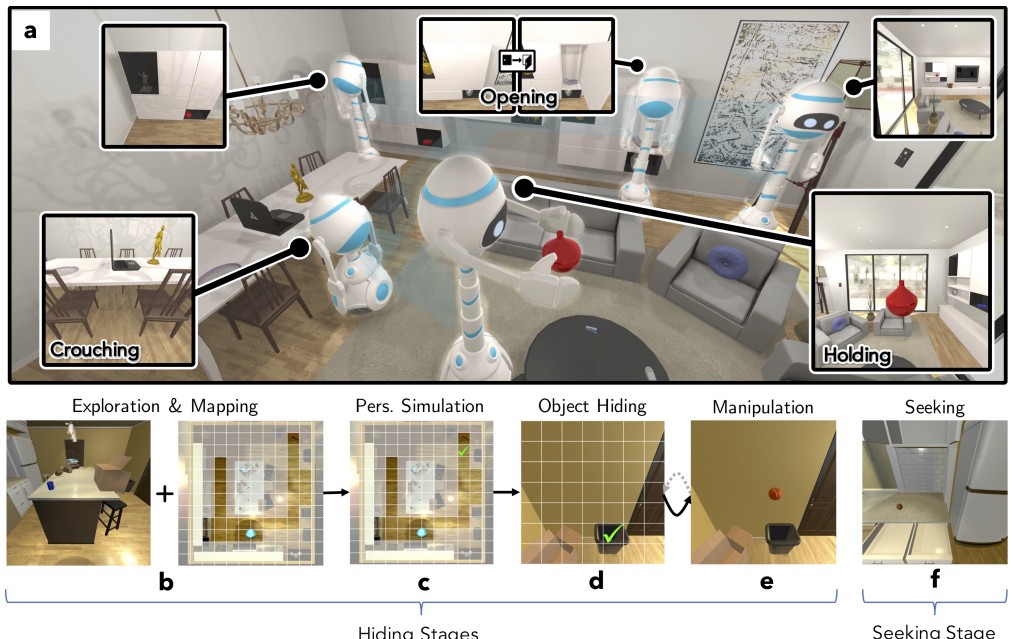

Figure 1: **A game of cache within AI2-THOR. a** Multiple views of a single agent who, over time, explores and interacts with objects. **b-f** The five consecutive stages of cache. In c the agent must choose where to hide the object at a high level, using its map of the environment to make this choice, while in d the agent must choose where to hide the object from a first person viewpoint. In e the object being manipulated is a tomato.

ment (Mnih et al., 2016) and self-supervised learning. After our agents are trained to play cache, we then probe how they have learned to represent their environment. To this end we distinguish two distinct categories of representations generated by our agents. The first, static image representations (SIRs), correspond to the output of a CNN applied to the agent's egocentric visual input. The second, dynamic image representations (DIRs), correspond to the output of the agents' RNN. While SIRs are timeless, operating only on single images, DIRs have the capacity to incorporate the agent's previous actions and observations.

Representation learning within the computer vision community is largely focused on developing SIRs whose quality is measured by their utility in downstream tasks (*e.g.* classification, depth-prediction, etc) (Zamir et al., 2018). Our first set of experiments show that our agents develop low-level visual understanding of individual images measured by their capacity to perform a collection of standard tasks from the computer vision literature, these tasks include pixel-to-pixel depth (Saxena et al., 2006) and surface normal (Fouhey et al., 2013) prediction, from a single image. While SIRs are clearly an important facet of representation learning, they are also definitionally unable to represent an environment as a whole: without the ability to integrate observations through time, a representation can only ever capture a single snapshot of space and time. To represent an environment holistically, we require DIRs. Unlike for SIRs, we are unaware of any well-established benchmarks for DIRs. In order to investigate what has been learned by our agent's DIRs we develop a suite of experiments loosely inspired by experiments performed on infants and young children. These experiments then demonstrate our agents' ability to integrate observations through time and understand spatial relationships between objects (Casasola et al., 2003), occlusion (Hespos et al., 2009), object permanence (Piaget, 1954), and seriation (Piaget, 1954) of free space.

It is important to stress that this work focuses on studying how play and interaction contribute to representation learning in artificial agents and not on developing a new, state-of-the-art, methodology for representation learning. Nevertheless, to better situate our results in context of existing work, we provide strong baselines in our experiments, *e.g.* in our low-level vision experiments we compare against a fully supervised model trained on ImageNet (Deng et al., 2009). Our results provide compelling evidence that: (a) on a suite of low level computer vision tasks within AI2-THOR, static representations learned by playing *cache* perform very competitively (and often outperform) strong unsupervised and fully supervised methods, (b) these static representations, trained using only synthetic images, obtain non-trivial transfer to downstream tasks using real-world images, (c)

unlike representations learned from datasets of single images, agents trained via embodied gameplay learn to integrate visual information through time, demonstrating an elementary understanding of free space, objects and their relationships, and (d) embodied gameplay provides a natural means by which to generate rich experiences for representation learning beyond random sampling and relatively simpler tasks like visual navigation.

In summary, we highlight the contributions: (1) Cache – we introduce cache within the AI2-THOR environment, an adversarial game which permits the study of representation learning in the context of interactive, visual, gameplay. (2) Cache agent – training agents to play Cache is non-trivial. We produce a strong Cache agent which integrates several methodological novelties (see, *e.g.*, perspective simulation and visual dynamics replay in Sec. 4) and even outperforms humans at hiding on training scenes (see Sec. 5). (3) Static and dynamic representation study – we provide comprehensive evaluations of how our Cache agent has learned to represent its environment providing insight into the advantages and current limitations of interactive-gameplay-based representation learning.

## 2  RELATED WORK

**Deep reinforcement learning for games.** As games provide an interactive environment that enable agents to receive observations, take actions, and receive rewards, they are a popular testbed for RL algorithms. Reinforcement learning has been studied in the context of numerous single, and multi, agent games such as Atari Breakout (Mnih et al., 2015), VizDoom (Lample & Chaplot, 2017), Go (Silver et al., 2017), StarCraft (Vinyals et al., 2019), Dota (Berner et al., 2019) and Hide-and-Seek (Baker et al., 2020). The goal of these works is proficiency: to create an agent which can achieve (super-) human performance with respect to the game's success criteria. In contrast, our goal is to understand how an agent has learned to represent its environment through gameplay and to show that such an agent's representations can be employed for downstream tasks beyond gameplay.

**Passive representation learning.** There is a large body of recent works that address the problem of representation learning from static images or videos. Image colorization (Zhang et al., 2016), ego-motion estimation (Agrawal et al., 2015), predicting image rotation (Gidaris et al., 2018), context prediction (Doersch et al., 2015), future frame prediction (Vondrick et al., 2016) and more recent contrastive learning based approaches (He et al., 2020; Chen et al., 2020b) are among the successful examples of passive representation learning. We refer to them as *passive* since the representations are learned from a fixed set of images or videos. Our approach in contrast is *interactive* in that the images observed during learning are decided by the actions of the agent.

**Interactive representation learning.** Learning representations in dynamic and interactive environments has been addressed in the literature as well. In the following we mention a few examples. Burda et al. (2019) explore curiosity-driven representation learning from a suite of games. Anand et al. (2019) address representation learning of the latent factors used for generating the interactive environment. Zhan et al. (2018) learn to improve exploration in video games by predicting the memory state. Ghosh et al. (2019) learn a functional representation for decision making as opposed to a representation for the observation space only. Whitney et al. (2020) simultaneously learn embeddings of state and action sequences. Jonschkowski & Brock (2015) learn a representation by measuring inconsistencies with a set of pre-defined priors. Pinto et al. (2016) learn visual representations by pushing and grasping objects using a robotics arm. The representations learned using these approaches are typically tested on tasks akin to tasks they were trained with (e.g., a different level of the same game). In contrast, we investigate whether cognitive primitives such as depth estimation and object permanence maybe be learned via gameplay in a visual dynamic environment.

## 3  PLAYING CACHE IN SIMULATION

We situate our agents within AI2-THOR, a simulated 3D environment of indoor living spaces within which multiple agents can navigate around and interact with objects (*e.g.* by picking up, placing, opening, closing, cutting, switching on, etc.). In past works, AI2-THOR has been leveraged to teach agents to *interact with their world* (Zhu et al., 2017; Hu et al., 2018; Huang et al., 2019; Gan et al., 2020b; Wortsman et al., 2019; Gordon et al., 2017), *interact with each other* (Jain et al., 2019; 2020) as well as *learn from these interactions* (Nagarajan & Grauman, 2020; Lohmann et al., 2020). AI2-THOR contains a total of 150 unique scenes equally distributed into five scene types: kitchens, living rooms, bedrooms, bathrooms, and foyers. We train our cache agents on a subset of the kitchen

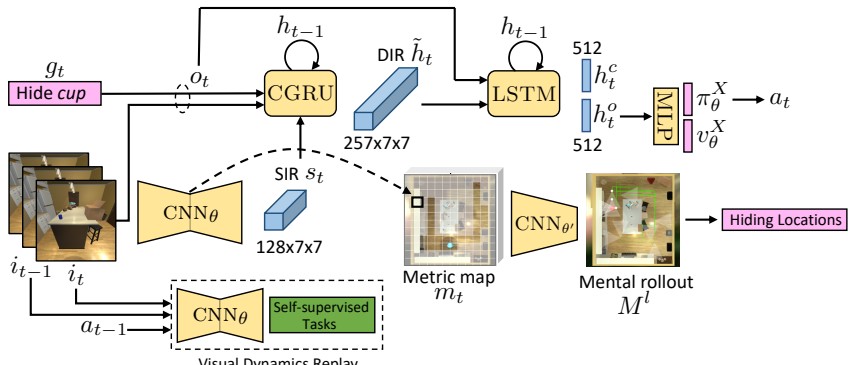

Figure 2: **Overview of the neural network architecture for the Cache agent.** The trainable components are shown in yellow, the inputs and outputs in pink, and the intermediate outputs in blue. Refer to text for details.

and living room scenes as these scenes are relatively large and often include many opportunities for interaction; but we use all scenes types across our suite of experiments. Details regarding scene splits across train, validation, and test for the different experiments can be found in Sec. A.1.

In a game of cache, two agents (a *hider* and a *seeker*) compete, with the hiding agent attempting to place a given object in the environment so that the seeking agent cannot find it. This game is zero-sum with the hiding agent winning if and only if the seeking agent cannot find the object. We partition the game of cache into five conceptually-distinct stages: exploration and mapping (E&M), perspective simulation (PS), object hiding (OH), object manipulation (OM), and seeking (S); see Figures 1b to 1f. A game of cache begins with the hiding agent exploring its environment and building an internal map corresponding to the locations it has visited (E&M). The agent then chooses globally, among the many locations it has visited, a location where it believes it can hide the object so that the seeker will not be able to find it (PS). After moving to this location the agent makes a local decision about where the object should be placed, e.g. behind a TV or inside a drawer (OH). The agent then performs the low-level manipulation task of moving the object in its hand to the desired location (OM), if this manipulation fails then the agent tries another local hiding location (OH). Finally the seeking agent searches for the hidden object (S). Pragmatically, these stages are distinguished by the actions available to the agent (see Table G.1 for a description of all 219 actions) and their reward structures (see Sec. C.5). For more details regarding these stages, see Sec. B.

## 4 LEARNING TO PLAY CACHE

In the following we provide an overview of our agents' neural network architecture as well as our training and inference pipelines. For space, comprehensive details are deferred Sec. D.

Our agents are parameterized using deep convolutional and recurrent neural networks. In what follows, we will use $\theta \in \Theta$ as a catch-all parameter representing the learnable parameters of our model. Our model architecture (see Fig 2) has five primary components: (1) a CNN that encodes input egocentric images from AI2-THOR into SIRs, (2) a RNN model that transforms input observations into DIRs, (3) multi-layer perceptrons (MLPs) applied to the DIR at a given timestep, to produce the actor and critic heads, (4) a perspective simulation module which evaluates potential object hiding places by simulating the seeker, and (5) a U-Net (Ronneberger et al., 2015) style decoder CNN used during visual dynamics replay (VDR), detailed below.

**Static representation architecture**. Given images of size $3 \times 224 \times 224$, we generate SIRs using a 13-layer U-Net style encoder CNN, $\text{CNN}_\theta$. $\text{CNN}_\theta$ downsamples input images by a factor of 32 and produces an our SIR of shape $128 \times 7 \times 7$.

**Dynamic representation architecture**. Unlike an SIR, DIRs combine current observations, historical information, and goal specifications. Our model has three recurrent components: a metric map, a 1-layer convolutional-GRU $\text{CGRU}_\theta$ (Cho et al., 2014; Kalchbrenner et al., 2016), and a 1-layer LSTM $\text{LSTM}_\theta$ (Hochreiter & Schmidhuber, 1997). At time $t \geq 0$, the environment provides the agent with observations $o_t$ including the agent's current egocentric image $i_t$ and goal $g_t$ (*e.g.* hide the cup). $\text{CNN}_\theta$ inputs $i_t$ to produce the SIR $s_t = \text{CNN}_\theta(i_t)$. Next the CGRU processes existing

information to produce the (grid) DIR $\widetilde{h}_t = \text{CGRU}_\theta(s_t, o_t, h_{t-1})$ of shape $257{\times}7{\times}7$. $\widetilde{h}_t$ is then processed by the LSTM $\text{LSTM}_\theta$ to produce the (flat) DIRs $(h_t^o, h_t^c) = \text{LSTM}_\theta(\widetilde{h}_t, o_t, h_{t-1})$ where $h_t^o, h_t^c$ are the so-called *output* and *cell* states of the LSTM with length 512. Finally the agents' metric map representation $m_{t-1}$ from time $t-1$ is updated by inserting a 128-dim tensor from $\text{CNN}_\theta$ into the position in the map corresponding to the agent's current position and rotation, to produce $m_t$. The historical information $h_t$ for the next step now includes $(\widetilde{h}_t, h_t^o, h_t^c, m_t)$.

**Actor-critic heads**. Our agents use an actor-critic style architecture, producing a policy and a value at each time $t$. As described in Sec. 3, there are five distinct stages in a game of cache (E&M, PS, OH, OM, S). Each stage has unique constraints on the space of available actions and different reward structures, and thus require distinct polices and value functions – obtained by distinct 2-layer MLPs taking as input the DIR $h_t^o$ described above.

**Perspective Simulation**. Evidence suggests that in humans, perspective taking, the ability to take on the viewpoint of another develops well after infancy (Trafton et al., 2006). While standard RNNs can perform sophisticated tasks, there is little evidence to suggest they can perform the multi-step if-then reasoning necessary in perspective taking. Inspired by AlphaGo's (Silver et al., 2017) intuition guided Monte-Carlo tree search and imagination-augmented RL agents (Racanière et al., 2017) we require, in the perspective simulation stage, that our hiding agent evaluate prospective hiding places by explicitly taking on the seeker's perspective using internal mental rollouts (see Fig. 3c).

To this end, the metric map is processed by a CNN to produce an internal map representation $M^1$, which is used to coarsely evaluate potential hiding locations, see Fig. 3b. For a subset of these locations $\ell$, $M^1$ is processed (along with $\ell$) to produce a new representation $M^\ell$. Intuitively, $M^\ell$ can be thought of as representing that agent's beliefs about what the environment would look like after having hidden the object at position $\ell$. A "mental" seeker agent defined using an LSTM (not shown in the model figure for simplicity) then moves about $M^\ell$ and obtains observations as slices of $M^\ell$, see Fig. 3c. The quality of a potential hiding location is then quantified by the number of steps taken by the "mental" seeking agent to find the hidden object. After scoring hiding locations in this way, the agent randomly samples a position to move to (beginning the OH-stage) with larger probability given to higher scoring positions.

**Training and losses.** Our agents are trained using the asynchronous advantage actor-critic (A3C) algorithm (Mnih et al., 2016) with generalized advantage estimation (GAE) (Schulman et al., 2015) and several (self) supervised losses. As we found it critical to producing high quality SIRs, we highlight our visual dynamics methodology below. See Sec.C for details on all losses.

**Visual dynamics replay.** During early experiments, we found that producing high quality SIRs using only existing reinforcement learning techniques to be infeasible as the produced gradients suffer from high variance and so signal struggles to propagate through the many convolutional layers. To correct this deficiency we introduce a technique we call a visual dynamics replay (VDR) inspired by experience replay (Mnih et al., 2015) and intrinsic motivation (Pathak et al., 2017). In VDR, agent state transitions produced during training are saved to a separate process where an encoder-decoder architecture (using $\text{CNN}_\theta$ as the encoder) is trained to predict a number of targets including forward dynamics, predicting an image from the antecedent image and action, and inverse dynamics, predicting which action produced the transition from one image to another. Given triplet $(i_0, a, i_1)$, where $i_0, i_1$ are the images before and after an agent took action $a$, we employ several self-supervised tasks including pixel-to-image predication. See Sec. C.2 for more details.

## 5  EXPERIMENTS

Our experiments are designed to address three questions: (1) "has our cache agent learned to proficiently hide and seek objects?", (2) "how do the SIRs learned by playing cache compare to those learned using standard supervised approaches and when training using other interactive tasks?", and (3) "has the cache agent learned to integrate observations over time to produce general DIR representations?". For training details, *e.g.* learning rates, discounting parameters, etc., see Sec. C.

**Playing cache.** Figures 3a to 3f qualitatively illustrate that our agents have learned to explore, navigate, provide intuitive evaluations of hiding spaces, and manipulate objects. Figures 4a and 4b, show quantitatively that hiders from later in training consistently chose hiding places that our seekers are less likely to find. Also, for hiding spots generated by human experts and by an automated brute force approach, seekers from later in training more frequently find the hidden objects. Interestingly,

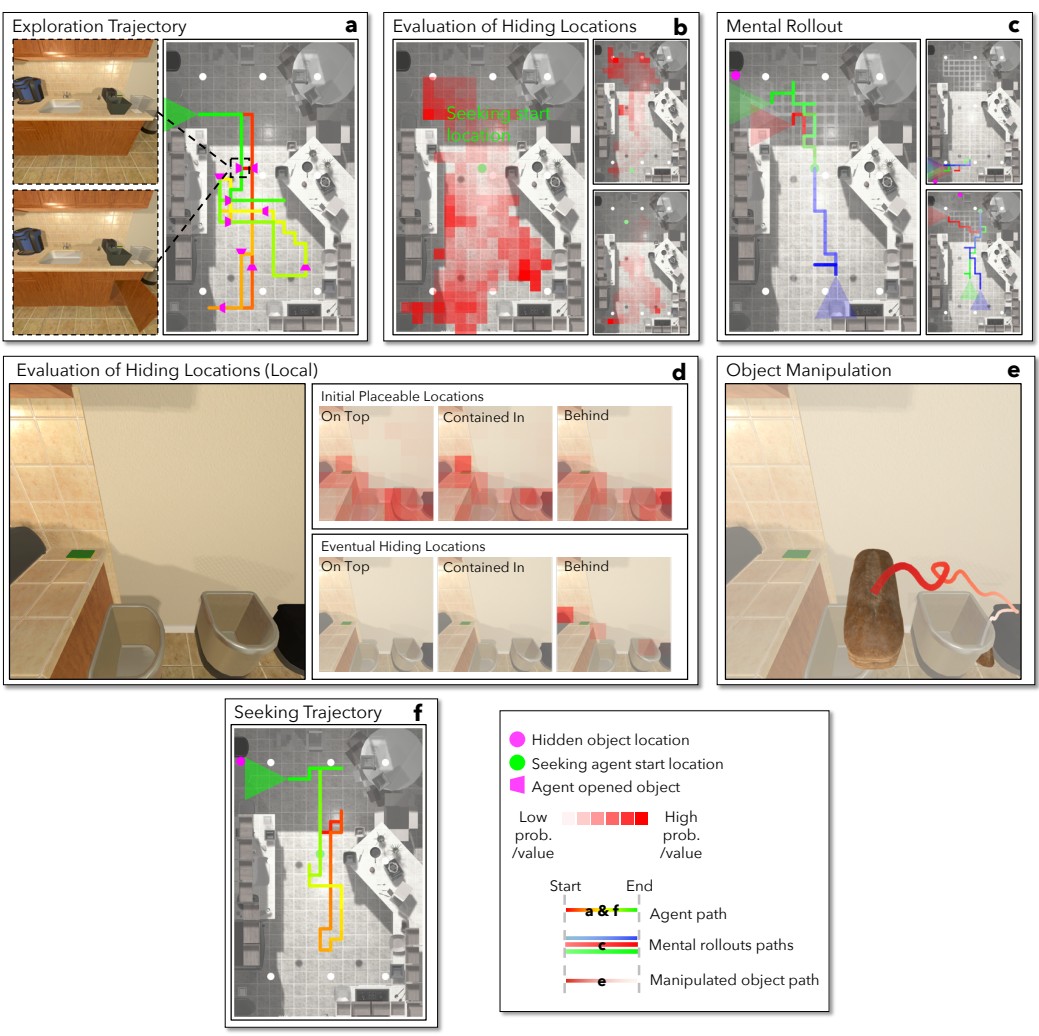

Figure 3: **Cache agent.** Actions and decisions made during a game of cache. **a** Hider's trajectory during exploration. Insets show first person views before and after an *open* action. **b** Heatmap displaying the agent's initial beliefs of high quality hiding locations. Insets show how this heatmap varies when the seeking agent's start position is changed. **c** Initial beliefs in b are refined by the agent performing explicit mental rollouts of the seeking agent. Three trajectories, stopped after 30 steps, are highlighted. Faint white lines are paths from other simulated trajectories. Insets show rollouts when varying the hiding location and agent starting location. **d** After choices made in b/c, a hiding location for the object must be chosen in pixel space. Top row shows the estimated probability that the object can be placed in a location. The bottom row shows the probability, averaged over the episode, of our agent trying to hide the object in a location. **e** The object's trajectory through space is shown as the agent places it between two trash cans. **f** Top down view of the seeker.

the seeker is able to locate objects hidden by humans and the brute force approach with a similar level of accuracy. Surprisingly, as shown in Table 4c, human seekers find objects hidden by humans and by our hiding agent at similarly high rates: on training scenes human seekers find our agent's objects 77% of the time and their fellow humans hidden objects 80% of the time, on test scenes humans are more successful at finding our agent's objects (90% success rate). Finally note that the (effectively random) $h_0$ agent hides objects generally very near its starting location and such positions are very frequently found by all agents. Somewhat surprisingly, the $s_{1e6}$ agent appears very slightly worse at finding such positions than the seeker $s_0$ agent. This can be explained by the $s_{1e6}$ agent having learned to not expect such an audacious hiding strategy.

**Static tasks.** We evaluate SIRs on a suite of standard computer vision targets such as monocular depth estimation by freezing the SIRs and training decoder models. These evaluations are performed on synthetic (AI2-THOR) data as well as natural images, which can be considered as out of domain.

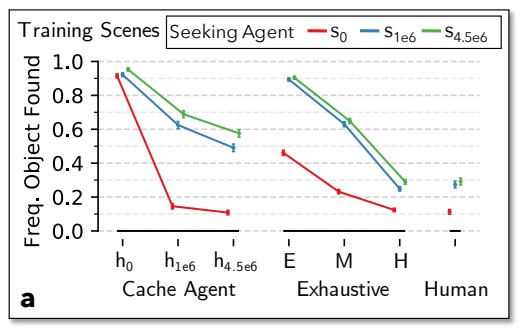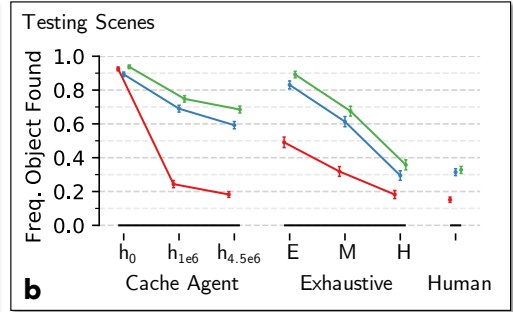

| Seeker | Hider | Split | Freq. Found | Seeker Steps |
|--------|-------|-------|-------------|--------------|
| $s_{4.5e6}$ | $h_{4.5e6}$ | Train | $0.44 \pm 0.01$ | $362 \pm 4$ |
| $s_{4.5e6}$ | Human | Train | $0.29 \pm 0.01$ | $406 \pm 4$ |
| Human | $h_{4.5e6}$ | Train | $0.77 \pm 0.03$ | $198 \pm 15$ |
| Human | Human | Train | $0.80 \pm 0.03$ | $204 \pm 16$ |
| $s_{4.5e6}$ | $h_{4.5e6}$ | Test | $0.64 \pm 0.01$ | $265 \pm 4$ |
| $s_{4.5e6}$ | Human | Test | $0.33 \pm 0.01$ | $386 \pm 4$ |
| Human | $h_{4.5e6}$ | Test | $0.90 \pm 0.02$ | $108 \pm 9$ |
| Human | Human | Test | $0.77 \pm 0.03$ | $155 \pm 12$ |

**c**

Figure 4: **Evaluating the cache agent. a,b** Frequency that seekers from three points during training (after 0, $1 \cdot 10^6$, and $4.5 \cdot 10^6$ training episodes) find objects hidden by: hiders from the same training points, an automated exhaustive approach ($E$=easy, $M$=medium, $H$=hard), and humans. Error bars denote 95% CIs. **c** Results of our best hider ($h_{4.5e6}$) and seeker ($s_{4.5e6}$) competing with humans. Here the mean frequency of finding the hidden object is reported (with 95% CIs) along with the mean number of steps the seeker took to find the hidden object (with 95% CIs).

*Predictions with synthetic images* - We use SIRs to predict geometry and appearance-based targets namely depth, surface normals, object class, object depth and object normals, as well as affordance-based targets namely traversable surfaces and object openability. On the test set, our (Cache) SIR consistently outperforms multiple baselines (Fig 5) including: *Auto Encoder* on AI2-THOR images, a *Navigation* agent within AI2-THOR, and *Classifier* – a CNN trained to classify objects in AI2-THOR images. Importantly, the Cache SIRs compare favorably (statistically indistinguishable performance for 4 tasks and statistically significant improvement for the remaining 2) to SIRs produced by a model trained on the 1.28 million hand-labelled images from ImageNet (Deng et al., 2009), the gold-standard for fully-supervised representation learning with CNNs. Note that our SIR is trained without any explicitly human labeled examples.

*Predictions with natural images* - As images from AI2-THOR are highly correlated and synthetic, we would not expect cache SIRs to transfer well to natural images. Despite this, Fig. 6 shows that the cache SIRs perform remarkably well: for SUN scene classif. (Xiao et al., 2010) as well as the NYU V2 depth (Nathan Silberman & Fergus, 2012) and walkability tasks (Mottaghi et al., 2016), the cache SIRs outperform baselines and even begin to approach the ImageNet SIR performance. This shows that there is strong promise in using gameplay as a method for developing SIRs. Increasing the diversity and fidelity of synthetic environments may allow us to further narrow this gap.

**Dynamic tasks.** While CNN models can perform surprisingly sophisticated reasoning on individual images, these representations are fundamentally timeless and cannot hope to model visual understanding requiring memory and intentionality. Through experiments loosely inspired by those from developmental psychology we show that, memory enhanced representations emerge within our agents' DIRs when playing cache. In the following experiments, we ensure that objects and scenes used to train the probing classifiers are distinct from those used during testing. For a description of all baseline models, see Sec G.

Developmental psychology has an extensive history of studying childrens' capacity to differentiate between an object being contained in, or behind, another object (Casasola et al., 2003). Following this line of work, our hider is made to move an object through space until the object is placed within, or behind, a receptacle object (see Fig. 7a). The DIR is then used to predict the relationship between

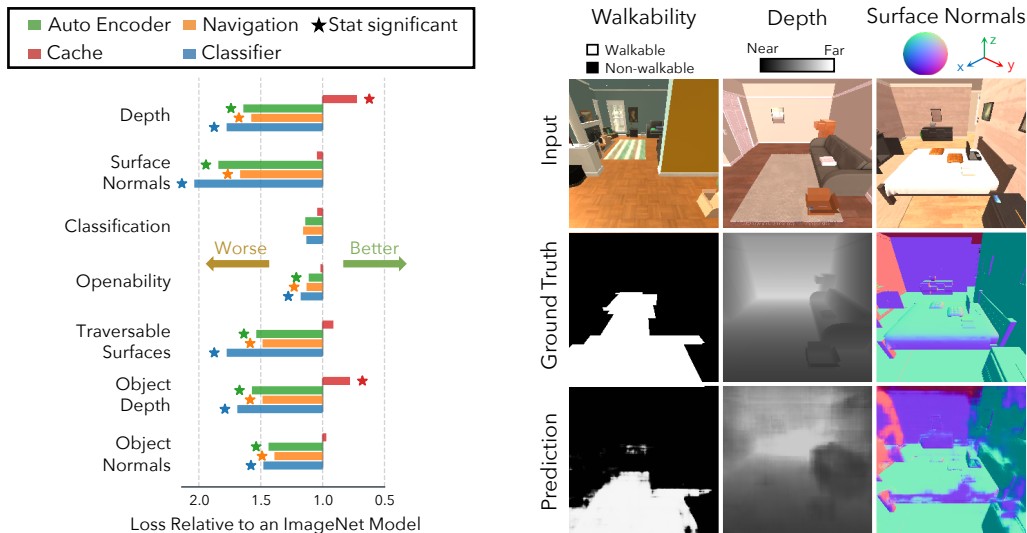

Figure 5: **SIR evaluation on synthetic images.** Left: test-set losses (averaged first within each scene) of the different models relative to the fully supervised baseline (ImageNet). Values less than 1 indicate performance better than that of the ImageNet SIR (stat. significance computed at a <0.01 level using a generalized linear mixed-model). Right: qualitative examples of geometric and affordance-based image predictions using SIR.

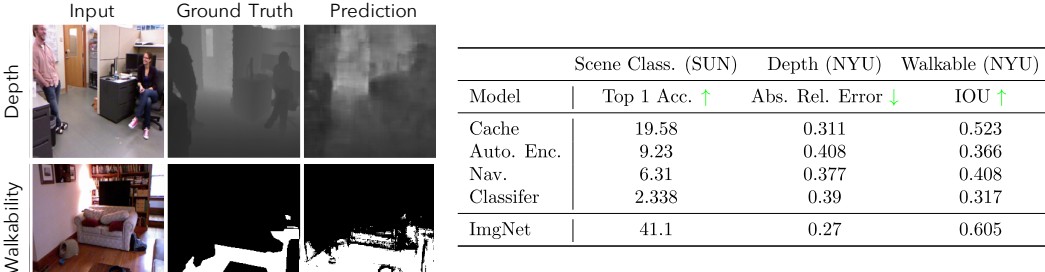

|  | Scene Class. (SUN) | Depth (NYU) | Walkable (NYU) |
| --- | --- | --- | --- |
| Model | Top 1 Acc. ↑ | Abs. Rel. Error ↓ | IOU ↑ |
| Cache | 19.58 | 0.311 | 0.523 |
| Auto. Enc. | 9.23 | 0.408 | 0.366 |
| Nav. | 6.31 | 0.377 | 0.408 |
| Classifer | 2.338 | 0.39 | 0.317 |
| ImgNet | 41.1 | 0.27 | 0.605 |

Figure 6: **SIR evaluation on natural images.** Quantitative and qualitative results for natural image tasks.

the object and the receptacle using a linear classifier. Cache DIRs dramatically outperform other methods obtaining 71.5% accuracy (next best baseline obtaining 63.4%).

Two and a half month old infants begin to appreciate object permanence (Aguiar & Baillargeon, 2002) and are surprised by impossible occlusion events highlighting the importance of permanence as a building block of higher level cognition, and begin acting on this information at around nine months of age (Hespos et al., 2009). Inspired by these studies, we first find that (Fig. 7b) a linear classifier trained on a cache DIR is able to infer the location of an object on a continuous path after it becomes fully occluded with substantially greater accuracy (64%) than the next best competing baseline (59.6%). Second (Fig. 7c), we test our agents' ability to remember and act on previously seen objects. We find that our seeker's DIR can be used to predict whether it has previously seen a hidden object after being forced to rotate away from it, having the object vanish, and taking four additional steps. Surprisingly we see that the seeker has statistically significantly larger odds of rotating back towards the object after seeing it than a control agent (mean difference in log-odds of rotating back towards the object vs. rotating in the other direction between treatment and control agents 0.08, 95% CI ±0.02) demonstrating its ability to act on its understanding of object permanence. Numerical markers denote the number of steps since the agent has seen the object.

Piaget argues that the capacity for seriation, ordering objects based on a common property, develops after age seven (Piaget, 1954). We cannot hope that our agents have learned a seven year old's understanding of seriation but we hypothesize that rudimentary seriation should be possible among properties central to cache. One such property is free space, the amount of area in front of the agent that could be occupied by the agent without collision. We place our agent randomly within the environment, rotate it in 90° increments (counter-) clockwise, and save SIRs and DIRs after each rotation (see Fig. 7d). We then perform a two-image seriation, namely to predict, from the representation at time $t$, whether the viewpoint at the previous timestep $t - 1$ contained more free

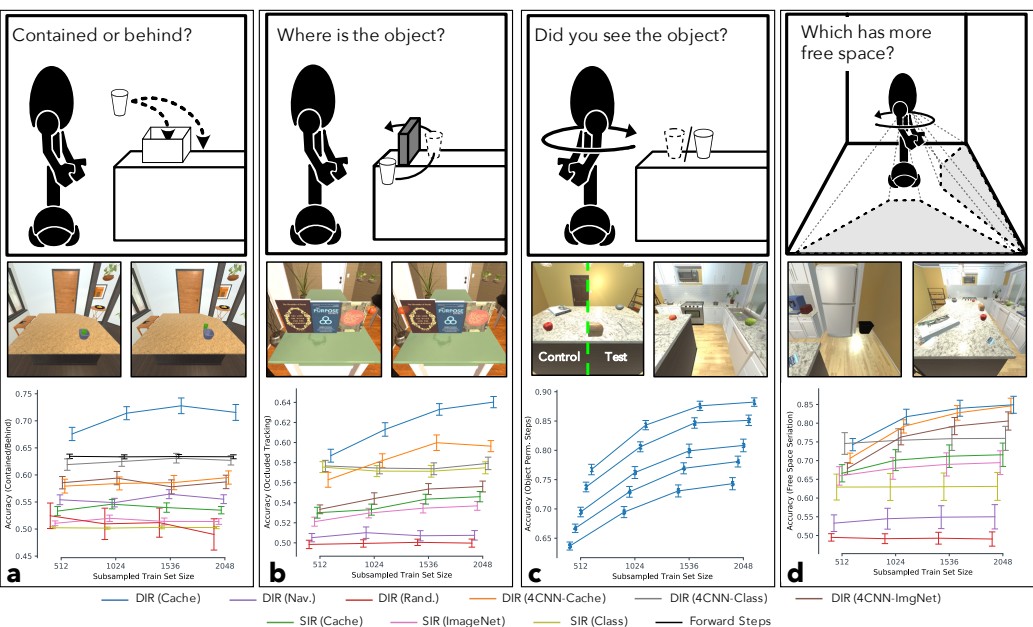

Figure 7: **DIR evaluation.** Schematic illustrations, example images from AI2-THOR and numerical results of our psychologically motivated experiments. All results are for the test set with 95% confidence intervals. See Sec. F for definitions of plot labels and Sec. 5 for a description of these experiments. Note: (i) X-coordinates in our line charts are offset slightly from one another so as to improve readability. (ii) Our plots show the performance of the representations as the amount of data used in training the probing linear classifier increases; but in the main text, we report only the values corresponding to the largest training set sizes considered.

space. Our cache DIR is able to perform this seriation task with accuracy significantly above chance (mean accuracy of 84.92%, 95% CI ±2.27%) and outperforms all baselines.

## 6 DISCUSSION

Modern paradigms for visual representation learning with static labelled and unlabelled datasets have led to startling performance on a large suite of visual tasks. While these methods have revolutionized computer vision, it is difficult to imagine that they alone will be sufficient to learn visual representations mimicking humans' capacity for visual understanding which is augmented by intent, memory, and an intuitive understanding of objects and their physical properties. Without such representations it seems doubtful that we will ever deliver one of the central promises of artificial intelligence, an embodied artificial agent who achieves high competency in an array of diverse, demanding, tasks. Given the recent advances in self-supervised and unsupervised learning approaches and progress made in developing visually rich, physics-based embodied AI environments, we believe that it is time for a paradigm shift via a move towards experiential, interactive, learning. Our experiments, in showing that rich interactive gameplay can produce representations which encode an understanding of containment, object permanence, seriation, and free space suggest that this direction is promising. Moreover, we have shown that experiments designed in the developmental psychology literature can be fruitfully translated to understand the growing competencies of artificial agents. Looking forward, real-world simulated environments like AI2-THOR have focused primarily on providing high-fidelity visual outputs in physics-enabled spaces with comparatively little effort being placed on developing other sensor modalities (*e.g.* sound, smell, texture, etc.). Recent work has, however, introduced new sensor modalities (namely audio) into simulated environments (Chen et al., 2020a; Gan et al., 2020a) and we suspect that the variety and richness of these observations will quickly grow within the coming years. The core focus of this work can easily be considered in the context of these emerging sensor modalities and we hope in future work to examine how, through representation and gameplay, an artificial agent may learn to combine these separate sensor streams to build a holistic, dynamic, representation of its environment.

### ACKNOWLEDGMENTS

We thank Chandra Bhagavatula, Jaemin Cho, Daniel King, Kyle Lo, Nicholas Lourie, Sarah Pratt, Vivek Ramanujan, Jordi Salvador, and Eli VanderBilt for their contributions in human evaluation. We also thank Oren Etzioni, David Forsyth, and Aude Oliva for their insightful comments on this project and Mrinal Kembhavi for inspiring this project.

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

# A  APPENDIX

## A.1  VIRTUAL ENVIRONMENT AND PREPROCESSING

While AI2-THOR can output images of varying dimensions and quality we find that simulating images of size $224{\times}224$ with the "Very Low" quality setting provides a balance of image fidelity and computational efficiency. AI2-THOR provides RGB images with the intensity of each channel encoded by an integer taking values between 0 and 255. Following standards used for natural images, we preprocess these images by first scaling the channels so that they are floating point numbers between 0 and 1 after which we normalize the R, G, and B channels with means 0.485, 0.456, 0.406, and standard deviations 0.229, 0.224, 0.225. Agents are constrained to lie on a grid with 0.25 meter increments and may face in one of four cardinal directions with $0°$ being north and $90°$ being east. The agent's camera is set to have a field of view of $90°$, is fixed at an angle $30°$ below the horizontal, and lies at a height of 1.5765 meters when standing and 0.9015 meters when crouching. Following conventions in computer graphics, the agent's position is parameterized a single $(x, y, z)$ coordinate triple with $x$ and $z$ corresponding to movement in the horizontal plane and with $y$ reserved for vertical movements. Other than when standing and crouching, our agent's vertical position is fixed. Letting $\mathbb{Z}$ denote the integers, an agent's position is uniquely identified by a *location tuple* $(x, z, r, s) \in (0.25{\cdot}\mathbb{Z}){\times}(0.25{\cdot}\mathbb{Z}){\times}\{0, 90, 180, 270\}{\times}\{0, 1\}$ where $x, z$ are as above, $r$ denotes the agent's rotation in degrees, and $s$ equals 1 if and only if the agent is standing. It will be useful in the following to be able to easily describe subcomponents of a location tuple, to this end we let $\mathrm{proj}_{-x} : \mathbb{R}^4 \to \mathbb{R}^3$ be the projection removing the first coordinate of a location tuple so that $\mathrm{proj}_{-x}(x, z, r, s) = (z, r, s)$, and similarly for $\mathrm{proj}_{-z}, \mathrm{proj}_{-r}, \mathrm{proj}_{-s}$. AI2-THOR contains a total of 150 scenes evenly grouped into five scene types, kitchens (1-30), living rooms (201-230), bedrooms (301-330), bathrooms (401-430), and foyers (501-530). Excluding foyers, which are reserved for our dynamic image representation experiments and used nowhere else, we consider the first 20 scenes of each scene type to be train scenes, the next five of each type to be validation scenes, and the last five of each type to be test scenes. When training our cache agents we only use kitchen and living room scenes as these scenes are relatively large and often include many opportunities for interaction.

# B  CACHE STAGES

In cache the *hiding agent*, or *hider*, attempts to place a *goal object* in a given AI2-THOR scene such that the *seeking agent*, or *seeker*, cannot find it. This goal object is one of five standard household items, see Fig. G.1, being either a loaf of bread, cup, knife, plunger, or tomato. Recall that we divide the game of cache into the five stages $\mathcal{S}^{\mathrm{stages}} = \{\text{E\&M, PS, OH, OM, S}\}$. The hider participates in the first four of these stages and the seeker in the last stage. Each of these stages is associated with a distinct discrete collection of possible actions, see Table G.1. When a stage $s \in \mathcal{S}^{\mathrm{stages}}$ is instantiated, so that the scene has been chosen and all goals specified, we call this instantiation an *s-episode*. During an *s*-episode, an agent takes actions, each taking a single unit of time called a *step*, until the goal state has been reached or a maximum number of steps, $N_s^{\mathrm{max\ steps}}$, has been taken. Episodes begin with time $t = 0$ and $t$ advances by 1 after every action regardless of whether or not that action is successful. Note that these episodes must be sequentially instantiated and we call a collection of sequentially instantiated episodes ending with an $S$-episode a *game of cache*.

In the following we describe these stages in detail along with the metrics we use to track the quality of the learned policies and assign rewards. In order to describe these metrics we require the following sets. First let $\mathcal{S}_{H,t}^{\mathrm{visited}}$ ($\mathcal{S}_{S,t}^{\mathrm{visited}}$) be the set of all location tuples visited by the hider (seeker) by time step $t$ of its episode. Similarly we let $\mathcal{S}_{H,t}^{\mathrm{opened}}$ ($\mathcal{S}_{S,t}^{\mathrm{opened}}$) be the set of all objects the hider (seeker) has opened by time step $t$. Finally we let $\mathcal{S}_{H,t}^{\mathrm{extrap.}}$ be the set of all location tuples that can, assuming the agent can move through objects, be reached by starting at some location in $\mathcal{S}_{H,t}^{\mathrm{visited}}$ and moving the agent ahead by 0.25m and then either staying put, moving left by 0.25m, or moving right 0.25m.

## B.1  EXPLORATION AND MAPPING (E&M)

A game of cache begins with an E&M episode. During this episode, the agent may traverse the scene and interact with objects while building a representation of its environment for up to $N_{E\&M}^{\mathrm{max\ steps}} = 200$ steps. In our implementation, this representation is a metric map, see the model architectures section below, which is then available in the subsequent PS- and OH-episodes. Let $\mathcal{S}^{\mathrm{reachable}}$ be the set of all feasible location tuples in a given episode and, similarly, let $N^{\mathrm{openable}}$ be total number of objects that can be opened. We evaluate the quality of exploration with three metrics:

$$Q^{\mathrm{coverage}} = |\,\mathrm{proj}_{-r}(\mathcal{S}_{H,200}^{\mathrm{visited}})|/|\,\mathrm{proj}_{-r}(\mathcal{S}^{\mathrm{reachable}})|,$$

$$Q^{\mathrm{coverage+}} = |\,\mathrm{proj}_{-r}(\mathcal{S}_{H,200}^{\mathrm{extrap.}}) \cap \mathrm{proj}_{-r}(\mathcal{S}^{\mathrm{reachable}})|/|\,\mathrm{proj}_{-r}(\mathcal{S}^{\mathrm{reachable}})|, \quad \text{and}$$

$$Q^{\mathrm{open\%}} = |\mathcal{S}_{H,200}^{\mathrm{opened}}|/N^{\mathrm{openable}}.$$

$Q^{\text{coverage}}$ and $Q^{\text{coverage+}}$ evaluate the agent's ability to traverse the entire scene while $Q^{\text{open%}}$ evaluates how frequently the agent interacts with its environment.

## B.2 PERSPECTIVE SIMULATION (PS)

After exploration the hider is allowed to consult the representation it has developed of its environment and must, from that representation, pick a single location tuple to which the agent will then be moved. As we do not give the agent a global reference frame it must choose this location tuple by specifying a location relative its final position after the E&M episode. As the hiding agent is not constrained to pick a valid location tuple, it may, for instance, choose a location that is outside of the scene or already occupied by an object, such a choice is considered a failed action. As the quality of a choice of location tuple in this episode is tied intimately with the outcome of the OH-episode we will measure the progress of training for both episodes using the same metrics described in the section below.

## B.3 OBJECT HIDING (OH)

In the OH-episode the agent has been moved to a location tuple determined in the PS-episode and is now holding the goal object. In this episode the agent is not allowed to perform any navigational actions beyond standing or crouching, but may interact with objects by opening or closing them. Most importantly, the agent must choose a location in its current view at which it should hide the item. To discretize the possible hiding locations recall that the images available to agents are of size $224\times224$. Viewing $C = \{1, ..., 224\}^2$ as a grid of pixel coordinates, let $(C_{ij})_{1\leq i,j\leq7}$ be a partitioning of $C$ into a $7\times7$ grid so that $C_{ij} = \{(32\cdot i + k, 32\cdot j + l) \mid 1 \leq l, k \leq 32\}$ is of size $32\times32$. When specifying a location in which to hide the object, the agent picks one of these grid locations along with one of the *positional keywords* in the set $\mathcal{S}^{\text{modality}} = \{\text{ONTOP}, \text{CONTAINEDIN}, \text{BEHIND}\}$. After choosing a location we begin an OM-episode, described below, during which the agent attempts to move the object through space and to the specified location. In order to reduce the complexity of specifying plausible object locations, potentially difficult when, *e.g.*, specifying a final location behind another object, we consider an object to have been successfully placed at the grid location $C_{ij}$, when using the CONTAINEDIN, BEHIND keywords, even in those cases that it moved to $C_{(i-1)j}, C_{(i+1)j}$ instead. If OM-episode is successful given this relaxation, all future actions specifying object locations are considered failures. If the OM-episode is unsuccessful the hiding agent may continue to specify locations, causing the item to be picked up and a new OM-episode to begin, until manipulation is successful. If the hiding agent fails to successfully place the object within $N_{OH}^{\text{max steps}} = 15$ steps the object is returned to its default position in the agent's hand and then simply dropped. Otherwise, the agent may continue to take other actions, closing a cabinet into which it has placed an object for instance, or specify that it is done by performing the `ReadyForSeeker` action.

Objectively quantifying a hiding location is difficult as the quality of a location is coupled with psychology and artifacts of perception. Indeed, the phenomenon of an object being lost "in plain sight" suggests that seemingly irrational hiding strategies may be highly successful. Nevertheless, we require some automated metric with which to measure the quality of a hidden object and, for this purpose, define the following three metrics. First we let $N^{\text{visible from}}$ be the number of feasible location tuples from which any portion of the goal object can be seen at any distance. To complement $N^{\text{visible from}}$ with a measure that takes the seeking agent's start location into account, we let $N^{\text{BFS steps}}$ be the number of steps it takes a breadth first search (BFS) agent, starting at the seeking agent's start location, to find the goal object. A single step for the BFS agent corresponds to teleporting to the location tuple closest to the seeker's start position that has not yet been visited by the BFS agent and checking if the goal object can be seen from any rotation, crouching or standing, from that location. The BFS agent is allowed to see through openable objects and the goal object is considered found if any portion of the object can be seen at any distance. If the BFS agent cannot find the object then $N^{\text{BFS steps}}$ equals the maximum number of steps, $|\text{proj}_{-r}(\text{proj}_{-s}(\mathcal{S}^{\text{reachable}}))|$, in which case we let $\delta^{\text{BFS found object}}$ equal 0, otherwise $\delta^{\text{BFS found object}}$ equals 1. Finally we also define the proportional versions of two of the above metrics.

$$Q^{\text{visible from%}} = N^{\text{visible from}}/|\mathcal{S}^{\text{reachable}}|, \quad \text{and}$$
$$Q^{\text{BFS steps%}} = N^{\text{BFS steps}}/|\text{proj}_{-r}(\text{proj}_{-s}(\mathcal{S}^{\text{reachable}}))|.$$

## B.4 OBJECT MANIPULATION (OM)

In the object manipulation episode the agent is set at a fixed location tuple holding the goal object and is given a grid location $1 \leq i, j \leq 7$ along with the positional keyword $\text{MODALITY} \in \mathcal{S}^{\text{modality}}$. During this episode the agent can interact with objects by opening them, move the object held in its hand in one of six axis-aligned directions, or drop the goal object. The object manipulation episode is considered successful if, after dropping the goal object, the goal object is in the correct grid position specified by $i, j$ and respects the constraints imposed by MODALITY. In particular, after the manipulator has dropped the object, we obtain from AI2-THOR the sets $\mathcal{S}^{\text{ONTOP pix.}} = \{$coordinates of pixels corresponding to the object that are visible from the agent's current viewpoint$\}$, $\mathcal{S}^{\text{BEHIND pix.}} = \{$pixel coordinates of the object that are visible only after making all other

objects in the scene invisible}, and the value $\delta^{\text{CONTAINEDIN}} \in \{0, 1\}$ which equals 1 if and only if the goal object is contained within another object. If $\delta^{\text{CONTAINEDIN}} = 1$, then we let $\mathcal{S}^{\text{CONTAINEDIN pix.}} = \mathcal{S}^{\text{ONTOP pix.}} \cup \mathcal{S}^{\text{BEHIND pix.}}$, otherwise $\mathcal{S}^{\text{CONTAINEDIN pix.}} = \emptyset$.

If $|\mathcal{S}^{\text{ONTOP pix.}}| + |\mathcal{S}^{\text{BEHIND pix.}}| \leq 10$, the episode is considered a failure. Otherwise, success depends on which MODALITY was specified and on the quantity $M = \max_{i,j} \max(|C_{i,j} \cap \mathcal{S}^{\text{ONTOP pix.}}|, |C_{ij} \cap \mathcal{S}^{\text{BEHIND pix.}}|, |C_{i,j} \cap \mathcal{S}^{\text{CONTAINEDIN pix.}}|)$, *i.e.* maximum number of pixels in any single modality and grid location.

If MODALITY = ONTOP then the episode was successful if $|C_{ij} \cap \mathcal{S}^{\text{ONTOP pix.}}| \geq 0.95 \cdot M$, i.e. if there are at least 95% as many pixels in the specified location and modality as in in the grid location and modality with the most of the object's pixels.

Similarly if MODALITY = BEHIND (or MODALITY = CONTAINEDIN) then the episode was successful if $|C_{ij} \cap \mathcal{S}^{\text{BEHIND pix.}}| \geq 0.95 \cdot M$ (or $|C_{ij} \cap \mathcal{S}^{\text{CONTAINEDIN pix.}}| \geq 0.95 \cdot M$).

To track the progress of this episode we simply record successes, namely we let $\delta^{\text{hit target}} \in \{0, 1\}$ equal 1 if and only if object manipulation was successful.

## B.5 SEEKING (S)

In the seeking episode the seeking agent is tasked with finding the object hidden by the hiding agent. The seeking agent begins from the same location tuple as the hiding agent began in the E&M-episode and given a budget of $N_S^{\text{max steps}} = 500$ steps before the episode is considered a failure. During these 500 steps the seeking agent may perform any of the actions allowed in the exploration and mapping episode along with one additional action, `ClaimVisible`. If the seeking agent performs the `ClaimVisible` action successfully the seeking agent has successfully found the goal object and the episode ends.

Just as when attempting to automatically measure the quality of a hiding location, developing objective measures of the quality of a seeking policy is challenging. As described previously, a seeking strategy that is highly successful against one hiding strategy may be fatally flawed when faced with another. To remove any dependence on a hiding agent obtained through reinforcement learning we generate, through an exhaustive, brute force approach, three collections of hiding locations for each scene $s$ and object $o$, $\mathcal{S}_{s,o}^{\text{easy pos.}}$, $\mathcal{S}_{s,o}^{\text{med. pos.}}$, and $\mathcal{S}_{s,o}^{\text{hard pos.}}$. To define these collections we use AI2-THOR to, for each goal object $o$ and scene $s$, generate a collection $L_{o,s} = \{\ell_{o,s,i} \mid 1 \leq i \leq N_{o,s}\}$ of $N_{o,s} \gg 0$ possible hiding locations. For each $\ell_{o,s,i}$ we compute $v_{o,s,i}$ which equals the $Q^{\text{visible from\%}}$ metric defined above for the given hiding location. Letting $V_{o,s} = \{v_{o,s,i} \mid 1 \leq i \leq N_{o,s}\}$, we consider a hiding position $\ell_{o,s,i}$ *hard* if $v_{o,s,i}$ is within the lower fifth percentile of $V_{o,s}$ values, *medium* if $v_{o,s,i} \leq 0.15$ and $v_{o,s,i}$ is between the fifth and 20th percentiles, and finally *easy* if it is neither hard nor medium difficulty. The set $\mathcal{S}_{s,o}^{\text{med. pos.}}$, and similarly for $\mathcal{S}_{s,o}^{\text{med. pos.}}$ and $\mathcal{S}_{s,o}^{\text{hard pos.}}$, is then a randomly sampled collection of, approximately, 20 easy locations from $L_{o,s}$. We then quantify the quality of a seeking policy by computing the success rate of the seeker when the hiding locations are restricted to those from each of the above three sets.

## C TRAINING DETAILS AND LOSSES

We train our cache agent using eight GPUs with one GPU reserved for running AI2-THOR processes, one reserved for VDR, and the other six dedicated to training with reinforcement and self-supervised learning. Each of the cache stages generates its own losses, detailed below, which are minimized by HOGWILD! (Recht et al., 2011) style unsynchronized gradient descent to train model parameters. Reinforcement learning in the E&M, OH, OM, and S stages is accomplished using the asynchronous advantage actor-critic (A3C) algorithm (Mnih et al., 2016) with generalized advantage estimation (GAE) (Schulman et al., 2015). For the A3C loss we let the discounting parameter $\gamma = 0.99$ (except for in the OH-stage where $\gamma = 0.8$), the entropy weight $\beta = 0.01$, and GAE parameter $\tau = 1$. We use the ADAM optimizer (Kingma & Ba, 2015) with AMSGrad (Reddi et al., 2018), moving average parameters $\beta_1 = 0.99, \beta_2 = 0.999$, a learning rate of $10^{-3}$ for VDR, and varying learning rates for the different cache stages ($10^{-4}$ for the E&M, OH, OM, and S stages, $5 \cdot 10^{-4}$ for the PS-stage).

In the following we give an overview various training details, namely: (1) data augmentation, (2) (self) supervised losses, (3) and reward structures for the E&M, OH, OM, and S stages necessary for computing the A3C loss.

### C.1 TRAINING DATA AUGMENTATION

To increase diversity of the images seen by our cache agents during training, we apply two forms of data augmentation to AI2-THOR.

**Random world rotation.** Recall that our AI2-THOR cache agents are constrained to $90°$ rotations and move on a 0.25m×0.25m grid. while this is beneficial in that it abstracts many of the technical complexities of movement in true robotics, it has the disadvantage of reducing the variety of images that can be seen by our agent. In particular, the default orientation of the agent is "axis aligned" such that it is generally moving parallel to walls and objects and thus it will not see objects and rooms from a large variety of orientations. To remedy this limitation while still maintaining grid-constrained movement, we, with a 50% probability at the start of every new game of cache, randomly rotate the agent by some degree in $\{0, \ldots, 359\}$. After this initial rotation the agent moves along a 0.25m×0.25m grid as usual but, because of this rotation, it will now observe objects at a variety of different orientations.

**Image augmentation.** To improve generalization, it is common with the computer vision literature to apply random transformations to images before giving them as input to neural network models for prediction. We follow this approach and, at the beginning of every game of cache, we randomly select:

(a) A resized crop function $C_{l,r,t,b}$ which, given an input image, crops $l$ pixels from the left, $r$ pixels from the right, $t$ pixels from the top, and $b$ pixels from the bottom, and then resizes the cropped image back to the start dimensions with bilinear sampling. In particular we select $(l, r, t, b) = (0, 0, 0, 0)$ with 0.5 probability and, otherwise, select $(l, r, t, b)$ uniformly at random from $\{1, \ldots, 10\}^4$.

(b) A color jitter function $J$. This function $J$ is chosen to vary the brightness, contrast, saturation, and hue of input images by small amounts.

(c) A rotation function $R_d$. Here $R_d$ rotates the input image about its center by $d$ degrees and we choose $d$ a random to equal 0 with probability 0.5 and, otherwise, sample $d$ uniformly at random from $\{-3, \ldots, 3\}$.

With the above functions in hand, we apply the composed transform $R_d \circ J \circ C_{l,r,t,b}$ to every image seen during the game of cache.

## C.2    VISUAL DYNAMICS REPLAY

Here we give an overview of our VDR training procedure. We direct anyone interested in exact reproduction of our VDR procedure to our code base. During training our agents save triplets $(i_0, a, i_1)$, where $i_0$ is an antecedent image, $a$ is an action taken (with associated success or failure signal), and $i_1$ is the subsequent image resulting from taking action $a$ when the agent's view was $i_0$, to a queue that is then periodically emptied and processed by the VDR thread. The VDR thread processes the triplets by saving them into a buffer, similar to an experience replay buffer (Mnih et al., 2015). To ensure that certain actions do not dominate the buffer we group and selectively subsample the triplets added to the buffer. Moreover, the buffer automatically removes triplets to ensure that triplets do not exceed some fixed age and that the buffer does not exceed its max size of approximately 20,000 triplets.

After processing incoming data and storing triplets in the dynamics buffer, the VDR thread then randomly partitions the buffer into batches of size 64 and, for each batch, computes the self-supervised losses detailed below and performs a single gradient step to minimize the sum of these losses. We call a single pass through all batches an epoch. VDR then alternates between processing incoming triplets, forming batches, and training for a single epoch.

The losses minimized in VDR include the following.

- Inverse dynamics loss: predicting $a$ from $i_0$ and $i_1$. From the VDR model we obtain, from $i_0$ and $i_1$, a vector $v^{\text{inv. dyn.}}$ of length $N^{\text{actions}}$= the total number of unique actions across all agents. By assigning each action $a$ a unique index, $\text{index}(a) \in \{1, ..., N^{\text{actions}}\}$, we then compute the inverse dynamics loss as the negative cross entropy between the point mass distribution $\delta^{\text{index}(a)}$ and $\text{softmax}(v^{\text{inv. dyn.}})$.

- Forward dynamics loss: predicting $i_1$ from $a$ and $i_0$. From the VDR model we obtain, from $i_0$ and $a$, a tensor $\widehat{i_1^{n \times n}} \in \mathbb{R}^{3 \times n \times n}$ for $n \in S = \{14, 28, 56, 112, 224\}$ where $\widehat{i_1^{n \times n}}$ represents the prediction of $i_1^{n \times n}$ which equals $i_1$ downsampled to have spatial resolution $n \times n$. Lett $\Delta_n$ be the, per pixel, $\ell_1$ error between $\widehat{i_1^{n \times n}}$ and $i_1^{n \times n}$, and let $\overline{\Delta}_n$ be its mean. We let the forward dynamics loss equal $\sum_{n \in S} \overline{\Delta}_n$. In practice we found it helpful to compute $\overline{\Delta}_n$ as a weighted average with larger weight given to areas of the image that have changed.

- Pixel difference loss: predicting which pixels in $i_1$ are different from those in $i_0$ from $i_0$ and $a$. Computed similarly to the forward dynamics loss using a tensor $T^{\text{pix. diff.}}$ returned from the VDR model except using a binary cross entropy loss to predict which pixels are different between $i_0$ and $i_1$.

- Pixel quality loss: predicting confidence of predictions of forward dynamics. From the VDR model we obtain a $1 \times 224 \times 224$ tensor $T^{\text{confidence}}$. We compute the pixel equality loss as the mean squared error between $T^{\text{confidence}}$ and $\Delta_{224}$.

- Action success loss: from $a$ and $i_0$ predict if the action would be successful. From the VDR model we obtain, from $a$ and $i_0$, $T^{\text{would succeed}}$. The action success loss is then computed as the binary cross entropy loss between $T^{\text{would succeed}}$ and the indicator $\delta^{\text{action did succeed}}$.

Note that we do not compute the forward dynamics loss and pixel quality loss on those triplets whose action $a$ (see table Table G.1 for all action types) is `RotateLeft`, `RotateRight`, `OpenAt`, `CloseObjects`, `PlaceAt`, `ReadyForSeeker`, or `ClaimVisible`. This is because such actions either cause very unpredictable changes (*e.g.* `RotateLeft` results in the agent looking at entirely separate portion of the scene) or no pixel changes (*e.g.* a `ReadyForSeeker` action simply ends an episode and changes nothing).

### C.3 TRAINING PERSPECTIVE SIMULATION

In this section, when we refer to a location tuple, we mean a location tuple without the standing component. Recall that in the PS-stage the agent must consult its map of the environment and, from this map, choose a location tuple where from which it will begin the OH-stage. In the following we describe our implementation of the PS-stage. We begin by describing the two-phase process by which a location tuple is chosen and then describe the losses we use to train this process. In the below description it will be helpful to define the set $\mathcal{S}^{\text{OH outcomes}} = \{\text{fail}\} \cup (\{0,1\} \times \{0,1,2\} \times \{1,...,7\}^2)$ where $\text{fail} \in \mathcal{S}^{\text{OH outcomes}}$ denotes the event in which the OH-episode did not successfully choose a hiding location (so the object was simply dropped) and $(s,m,i,j) \in \mathcal{S}^{\text{OH outcomes}}$ denotes the event in which, during the OH-stage, the agent successfully hid the object and

- the hiding agent was standing when it hid the object if $s = 1$ and was otherwise crouching,
- the hiding agent hid the object with a `PlaceAt|m,i,j` action.

For simplicity, we will say *an object was hidden at* $(\ell, e)$ to mean that, during the OH-stage, the agent was located at $\ell$ and $e \in \mathcal{S}^{\text{OH outcomes}}$ occurred. We now describe the two phase process for choosing a location tuple.

1. Initial evaluation (recall Fig, 3b). The agent assigns a score to all location tuples with the goal of highlighting locations that deserve further evaluation. This score, for a particular location tuple $\ell$, is computed as the weighted average

$$\sum_{e \in \mathcal{S}^{\text{OH outcomes}}} \text{softmax}(P^\ell)_e \cdot V_e^\ell$$

   where $P^\ell$ and $V^\ell$ are generated by the agent and are described below.

2. Mental rollout (recall Fig. 3c). The agent begins by picking the five highest scoring location tuples $\{\ell_1, ..., \ell_5\}$ from the previous phase along with five randomly selected location tuples $\{\ell_6, ..., \ell_{10}\}$. For each $1 \le i \le 10$, the seeking agent simulates the outcome of running the OH-stage with the agent at at location $\ell_i$ three times. Namely the agent iteratively samples three distinct elements $e_{i1}, e_{i2}, e_{i3} \in \mathcal{S}^{\text{OH outcomes}}$ with the probability of sampling $e_{ij} \in \mathcal{S}^{\text{OH outcomes}}$ at every iteration being proportional to $\text{softmax}(P^{\ell_i})_{e_{ij}}$. Then, for each $1 \le i \le 10$ and $1 \le j \le 3$, the agent simulates 50 rollout trajectories $(t_{ijk})_{k=1}^{50}$ of the seeking agent in the hypothetical setting in which the object was hidden at $(\ell_i, e_{ij})$. Let $\mu_{ij} = \frac{1}{50}\sum_{k=1}^{50} \#\text{steps}(t_{ijk})$ be the average number of steps taken in the simulated seeker trajectories $(t_{ijk})_{k=1}^{50}$. Straightforward analysis, using the probabilities defined in $\text{softmax}(P^{\ell_i})$, shows that there is closed-form solution for weights $w_{ij}$ so that $\mu_i = \sum_{j=1}^3 w_{ij}\mu_{ij}$ is an unbiased estimator of the expected mental rollout path length when hiding at $(\ell_i, e)$ and $e$ is selected from $\mathcal{S}^{\text{OH outcomes}}$ with probability $\text{softmax}(P^{\ell_i})_e$.

   After computing $\mu = (\mu_1, ..., \mu_{10})$, the location tuple $\ell_i$ where the actual OH-episode will begin is chosen by sampling $i$ from a $\text{Multinomial}(\text{softmax}(0.04 \cdot (\mu_1, ..., \mu_n)))$ distribution.

We now describe the losses minimized so that we obtain high quality initial evaluations of location tuples and mental rollouts of seeker trajectories.

Given a tensor $M$ encoding the agent's map of the environment (see the Metric mapping section below), the perspective simulation model produces an intermediate representation of this map $M^1$. From $M^1$ it generates two tensors $P^\ell$ and $V^\ell$ of size $2 \times 3 \times 7 \times 7$ as well as two values $p_{\text{none}}^\ell, v_{\text{none}}^\ell \in \mathcal{R}$. For simplicity we will abuse notation and let $P_{\text{none}}^\ell = p_{\text{none}}^\ell$ and $V_{\text{none}}^\ell = v_{\text{none}}^\ell$. Now $\text{softmax}(P^\ell)_e$ is a prediction of the probability that, were the agent to choose to hide the object at location $\ell$, the event $e \in \mathcal{S}^{\text{OH outcomes}}$ would occur. This tensor $\text{softmax}(P^\ell)$ is supervised using a cross-entropy loss with the hiding location chosen during the OH-stage. The entries of the tensor $V^\ell$ then represent an estimate of the quality of a hiding location. The collection of tensors $V^\ell$ is supervised as follows. Let $1 \le i \ne i' \le 10$ and $1 \le j \ne j' \le 3$. Now let $\mathcal{L}_{ij,i'j'}^V$ be the binary cross entropy loss between $V_{e_{ij}}^{\ell_i} - V_{e_{i'j'}}^{\ell_{i'}}$ and $\text{sign}(\mu_{ij} - \mu_{i'j'})$. Note that minimizing $\mathcal{L}_{ij,i'j'}^V$ encourages $V_{e_{ij}}^{\ell_i}$ to

be larger than $V_{e_{i'j'}}^{\ell_{i'}}$ if and only if $\mu_{ij} > \mu_{i'j'}$, namely in those settings where mental rollouts suggest that $(\ell_i, e_{ij})$ is a better hiding location than $(\ell_{i'}, e_{i'j'})$. We thus let our loss supervising the tensors $V^\ell$ equal the average $\mathcal{L}^V = \binom{30}{2}^{-1} \mathrm{sum}(\{\mathcal{L}_{ij,i'j'}^V \mid 1 \le i \le i' \le 10,\ 1 \le j \le j' \le 3,\ (i,j) \ne (i',j')\})$.

Now we discuss losses used to guide learning to produce mental rollouts of seeker trajectories. Suppose that a full game of cache has taken place. We begin by collecting several pieces of data from the game.

1. The location tuple $\ell$ during the PS-episode and the event $e \in \mathcal{S}^{\text{OH outcomes}}$ that occurred during the OH-stage.

2. The trajectory $(t_i)_{i=1}^n$ (ordered set of location tuples visited) by the seeker.

3. A $\{0, 1\}$-valued vector $v^{\text{would succeed}}$ with $i$th entry equalling 1 if and only if the seeker would have been able to successfully claim the object was visible at the $i$th step of its trajectory.

4. The actions $a_i$ taken by the seeking agent taken at step $i$ along with their success signal $s_i$.

5. The policy, probability distribution over actions, $\pi_i$ of the seeker at step $i$.

The hiding agent then forms, from the intermediary representation $M^1$ and pair $(\ell, e)$, a hiding-place-conditional representation $M^{\ell,e}$. Using this conditional representation as a "mental environment," the agent then simulates an internal seeker who followed the same trajectory $(t_i)_{i=1}^n$ and took the same actions $(a_i)_{i=1}^n$ as the true seeking agent. At every time step $i$ the hiding agent generates a mental policy $\widetilde{\pi}_i$ for the simulated seeker along with a vector of probabilities $\widehat{s}_i$ where $\widehat{s}_{ij}$ denotes the predicted probability that a seeking agent, at the location $t_i$, taking action with index $j$ would succeed in doing so. For every $i$, the mental policy $\pi_i$ is then trained, using a cross entropy loss, to mimic the true seeker policy $\pi_i$. Similarly at every step, the predictions $\widehat{s}_{ij}$ of whether actions would, or would no succeed, is supervised using a binary cross entropy loss with $s_i$ if $j$ is the index of action $a_i$. Furthermore each $\widehat{s}_{ij}$ corresponding to the `ClaimVisible` action is supervised binary cross entropy with $v^{\text{would succeed}}$.

## C.4 Additional losses

To improve training sample efficiency we augment the usual A3C losses, described below, by also computing several other losses, these losses are averaged across timesteps within each stage and then summed together along with any other losses, e.g. the A3C loss, and gradients are computed for this total loss before being synchronized with the shared model. As there are a number of additional losses, we will not describe the full mathematical details of these losses and instead describe their intent. Let $s \in \mathcal{S}^{\text{stages}}$, $t \ge 0$ be the current time step in $s$-episode, and $i_t$ be the image observed at that time step, if applicable.

Other than the A3C loss, there are two additional losses shared among the {E&M, OH, OM, S} stages described below:

1. Encouraging understanding the effect of actions: we would like for the DIR representation learned by our agents to be self-predictive, namely we would like the DIR to encapsulate how the environment will change when acted on by the agent. To this end, recall the notation from Section 4 where $s_t$ represented the $128 \times 7 \times 7$ SIR output from $\text{CNN}_\theta$ at time $t$ and $h_t^o$ represents the 512-dimensional hidden output vector created generated by $textsclstm_\theta$ at time $t$. Moreover let $a_t$ be the action taken by the agent at time $t$. To encourage self-predictivity, we then define two learnable embedding functions $E_\theta^1$ and $E_\theta^2$ and define a loss which is minimized when the dot product $E_\theta^1(s_{t+1}) \odot E_\theta^2(h_t^o, a_t)$ is large while $E_\theta^1(s_i) \odot E_\theta^2(h_t^o, a_t)$ is small for all $i \ne t$. Minimizing this loss encourages $h_t^o$ to contain enough information to be able to distinguish the true next visual observation $s_{t+1}$ from all other observations seen during the episode. Moreover it encourages the SIR $s_{t+1}$ to contain information that will make it readily distinguishable from other the SIRs of other timesteps.

2. Localizing changes in input images: similarly to the above, we would like our agent to be able to localize where changes have occurred in input images. As described in the neural models section below, at every time step the agent generates a $1 \times 7 \times 7$ tensor $W^{s,t}$ where $\mathrm{softmax}(W^{s,t})_{1,i,j}$ should is encouraged to be nonzero only if the pixels corresponding to $C_{ij}$ have changed from the previous to the current frame. To supervise $W^{s,t}$ we first generate the $1 \times 7 \times 7$, $\{0, 1\}$-valued, tensor $D$ where $D_{1,i,j}$ equals one if and only if the pointwise difference between the encodings of $i_t$ and $i_{t-1}$ generated by the first convolutional block of $\text{CNN}_\theta$, has 10% of the spatial locations corresponding to the $i, j$ grid location with $\ell_2$-norm greater than $10^{-3}$. We then form the loss

$$\mathcal{L} = -\log\left(\sum_{i=1}^7 \sum_{j=1}^7 \mathrm{softmax}(W^{s,t})_{1,i,j} \cdot D_{1,i,j}\right).$$

Note that minimizing $\mathcal{L}$ encourages the sum of the entries in $\mathrm{softmax}(W^{s,t})$ corresponding to entries in $D$ equalling one to be large.

Beyond the above losses shared among all of {E&M, OH, OM, S}, we also include several stage-specific losses described below.

**OH-specific losses**.

1. Avoiding early `PlaceAt` modality collapse: early in training the hiding agent makes many failed `PlaceAt` actions, as a large number of these actions are impossible for a given viewpoint, incurring a substantial penalty for these repeated failures. Because of this, we found that, in this stage, the agent often would often quickly learn to only use the ONTOP modality of the `PlaceAt` as this is the modality is, generally, the easiest for the manipulator to successfully complete. To prevent this policy collapse, we introduce a loss that encourages the agent to maintain probability mass on `PlaceAt` actions with CONTAINEDIN and BEHIND modalities. This is similar to the standard loss in A3C encouraging higher entropy in an agent's policy. This loss is removed after 5,000 episodes of training.

2. Avoiding early `PlaceAt` spatial collapse: similarly as above, we sometimes found that the agent's policy would concentrate on `PlaceAt` actions at a single grid position $i, j$ early in training. To prevent this, we create a loss encouraging the probability mass over `PlaceAt` to be similar to a prediction of which `PlaceAt` actions would be successful coming from the manipulator model (described below). As above, this loss is removed after 5,000 episodes of training.

3. Recalling `PlaceAt` actions: to encourage the agent to remember where it attempted to place the object and where, on those attempts, the object ended we designed two losses penalizing the agent for failing to accurate make such predictions correctly.

4. Closing objects: after placing an object so that is in the CONTAINEDIN modality we design a loss encouraging the agent to attempt closing objects.

**OM-specific losses**.

1. Better sample efficiency: inspired by hindsight experience replay (Andrychowicz et al., 2017), we design a loss that rephrases failed trajectories made during an OH-episode, i.e. a trajectory that placed the object in a location that was not specified as the goal location, into a successful trajectory where the goal was whichever location the object actually was placed in.

2. Opening objects before placing things into them: we designed a loss that encouraged the agent, during the OM-episode, to try opening an object whenever the goal location for the object was in a CONTAINEDIN modality.

3. Predicting if a goal object location was attainable: for better sample efficiency and for use during other stages, we encourage the agent to predict whether or not it will be able to manipulate the current object in such a way that it will reach the given goal object location. In particular, assume that the agent was directed to place the object in location $(m, i, j)$. The hiding agent then produces, at every step, a $3 \times 7 \times 7$ tensor $T^{\text{hittable, t}}$ and, at the end of the episode, computes an average binary cross entropy loss between the entries $T^{\text{hittable, t}}_{m,i,j}$ and the indicator $\delta^{\text{OM-episode success}}$.

**PS-specific losses**. See the section on training perspective simulation above.

**S-specific losses**.

1. Better sample efficiency through self-supervised imitation learning: using the mental representation of the environment created by hiding agent in the PS-stage we compute a shortest path from the seeking agent's current location to a location from which, in this mental representation, the object is visible and then encourage the seeking agent to follow this path.

2. Reducing false claim visible actions: at every step made by the seeker we encourage (or discourage) the agent from taking the `ClaimVisible` action based on whether or not the goal object is currently visible to the agent.

## C.5 REWARD STRUCTURE FOR REINFORCEMENT LEARNING

In order to compute the A3C loss we must define, for each stage $s \in \mathcal{S}^{\text{stages}}$, a reward structure, that is, a function which assigns a reward $r \in \mathbb{R}$ to every *state-action-state triplet* $(s, a, s')$, where $s$ denotes a possible agent state and $a$ an action which, when taken from state $s$, resulted in the agent moving into state $s'$. When applicable and unless otherwise stated, agents obtain a penalty of $r_{\text{step}} = -0.01$ on every step to discourage long trajectories and an additional penalty of $r_{\text{fail}} = -0.02$ whenever an action fails (for instance, when running into a wall). For clarity of exposition we will define the reward structure by defining $r$ in the case that agent starts at a generic state $s$ at time $t \geq 0$ and performs action $a$ resulting in a transition to state $s'$. *E&M reward structure.* During exploration and mapping, we would like to encourage the agent to comprehensively explore the environment, including interacting with and opening objects if possible, with particular attention being paid

to areas that would make good hiding locations. To this end, but for $r_{\text{E\&M, hide},t}$ described below, we define the E&M reward structure in Algorithm 1. Now for $0 \le s \le t+1$, at time $s$ let $m_s$ be the metric map generated by the agent, $\ell_s$ be the location tuple of the agent, and let $v_s$ be the value of the position $\ell_s$ when evaluating $m_s$ using the perspective simulation module. Now letting $\texttt{unseen}_s = \delta\{$with action $a_s$ the agent opened an object, or entered a location, for the first time$\}$, we define

$$\overline{v}_t = (\sum_{s=0}^{t} v_s \cdot \texttt{unseen}_s)/(\sum_{s=0}^{t} \texttt{unseen}_s)$$

so that $\overline{v}_t$ captures an average quality of unique states the agent has visited up to and including time $t$. We then define the reward

$$r_{\text{E\&M, hide},t} = 0.2 \cdot \min(1, \max(0, v_{t+1} - \overline{v}_t)) \cdot \texttt{unseen}_{t+1}$$

Note that $r_{\text{E\&M, hide},t}$ will be 0 if the agent has not opened a new object or entered a previously unvisited location tuple and, otherwise, will only be large if the perspective simulation module considers the current state to be substantially more valuable than previously visited states. Thus $r_{\text{E\&M, hide},t}$ provides a signal through which the exploration agent may learn to visit states that the perspective simulation module considers good hiding locations.

**OH reward structure.** Except for the definition of $r_{\text{OH, percentile}}$, the reward structure for the OH episode is defined in Algorithm 2. In the below we define and motivate $r_{\text{OH, percentile}}$. Suppose that the during the OH episode the hiding agent has successfully completed a `PlaceAt` action, we would like the reward given to the agent after its final action to reflect the quality of the hiding location chosen and thus a natural reward for this final action is number of steps taken by the seeker to find the object $N^{\text{S steps}}$ normalized by the maximum number of steps allotted to the seeking agent (500), i.e. the reward

$$r_{\text{S steps}} = N^{\text{S steps}}/500.$$

Unfortunately $r_{\text{S steps}}$ fails in two respects.

1. The reward $r_{\text{S steps}}$ suffers from very high variance. As an example, consider the case in which the object is hidden directly behind the seeker. If the seeker performs two `RotateRight` actions it will immediately find the object but, if it instead chooses to search the area ahead of itself then it may be many steps before it backtracks to its starting position and finds the object. In such a case the reward $r_{\text{S steps}}$ may be very large or very small with high probability.

2. The number of steps required to find the object is not only determined by the quality of the hiding place chosen in the OH episode but also by the topology of the scene and the location tuple chosen in the perspective simulation episode. Namely, in a large, maze-like, room the OH episode may terminate with the object in the open from the hiding agent's perspective (a poor hiding place) but still require many steps from the seeker to find the object as the location tuple chosen in perspective simulation was far away from the seeker's starting position.

This high variance and confounding with the quality of perspective simulation results in $r_{\text{S steps}}$ providing a poor signal as to where an object should be hidden in the OH episode. While the high variance of the reward is intrinsic to the game of cache, especially so in the case of randomly initialized agents, we can partially address the second issue above by considering a transformation of $r_{\text{S steps}}$ defined as follows. First we generate 100 sample seeker trajectories $(s_i)_{i=0}^{100}$ when simulating hiding the object at the location chosen during the PS-stage; see the training perspective taking section above for details as to how these trajectories are generated. Letting $\ell_i$ be the length of $s_i$, i.e. the number of steps the seeker took to find the goal object in that mental rollout, we then define

$$r_{\text{OH, percentile}} = -1 + 2 \cdot \frac{1}{100} \sum_{i=1}^{100} \delta^{[\ell_i \le N^{\text{S steps}}]}.$$

Assuming these mental rollouts are of reasonable quality, $r_{\text{OH, percentile}}$ naturally calibrates the reward for the topology and size of the scene. This helps disentangle the final reward in the OH episode from contributions made purely during the PS episode.

**OM reward structure.** The OM episode runs for a maximum of 50 steps and has the reward structure for this episode is outlined in Algorithm 3. Broadly speaking, we reward the agent for dropping the goal object near the target location with proximity measured in the image space and for placing the object in a location respecting the target modality among ONTOP, CONTAINEDIN, and BEHIND. We found it especially important to give the agent a large negative reward whenever it placed the object outside of the agent's viewport.

**S reward structure.** The reward structure of the seeking episode is simple and encodes our desire for the agent to locate the object in as short as time as is possible without having a large false positive rate, that is without having many instances in which the agent claims to be able to see the object when the object is not actually visible. This reward structure is given in Algorithm 4.

C.6    PRETRAINING THE E&M AND OM STAGES

As the ability to navigate about a room and perform object manipulation are basic competencies for the hiding agent in cache, we pretrain our hiding agent on the E&M and OM-stages until it attains reasonable proficiency. The reward structure when training the E&M-stage was essentially identical to that from Algorithm 1 but without the $r_{\text{E\&M,n hide},t}$ reward, this trained for approximately 85,000 episodes. To pretrain the OM-stages we generated, across AI2-THOR scenes, a collection of plausible locations the objects could be placed when at location tuples, and then trained the hiding agent to attempt placing objects in these locations using the reward structure in Algorithm 3; the manipulator agent was pretrained for approximately 1.8 million episodes. We ran this pretraining procedure for approximately three days.

# D    NEURAL MODELS

Here we give an overview of the neural network architectures we use to parameterize our agents. As these architectures can be complex, we do not attempt to make this description sufficiently comprehensive to exactly reproduce our results; we instead direct any parties interested in extract reproduction to our code base.

Our neural network architecture contains components that are shared among both the hider and seeker as well as components that are unique to individual stages. Recall from 4 that shared components include the image processing CNN $\text{CNN}_\theta$, the recurrent networks $\text{CGRU})\theta$ and $\text{LSTM}_\theta$ and metric mapping components. Previously unmentioned shared components include embeddings for actions, goal objects, and agent standing state. Unshared components include, for example, the MLPs defining actor-critic heads.

D.1    INFORMATION FLOW TO POLICY AND VALUE

Here we describe how, during the E&M-, OH–, OM-, or S-episode, data flows through our agents' neural architecture to produce the agent's policy and value of the current state.

Suppose the agent has received input corresponding to timestep $t \geq 0$ in stage $s \in \mathcal{S}^{\text{stages}}$ and must take an action. This input comprises the following.

- A $3 \times 224 \times 224$ (channels $\times$ height $\times$ width) tensor corresponding to the image of the agents' current view. This tensor is processed $\text{CNN}_\theta$ to produce an SIR $s_t$ of shape $128 \times 7 \times 7$.

- A non-negative integer specifying the agent's last action (including a 'none' action when $t = 0$) along with a 'success' or 'failure' signal indicating whether the agent's last action succeeded. This pair of inputs is used to index into an embedding matrix resulting in a 16-dimensional embedding vector.

- A non-negative integer specifying whether or not the agent is standing. This integer is also used to index into an embedding matrix resulting in a 4-dimensional embedding

- A non-negative integer specifying the goal object (i.e. object to hide or object to seek). This integer is also used to index into an embedding matrix resulting in a 16-dimensional embedding vector.

- The $257 \times 7 \times 7$-dimensional hidden state $\widetilde{h}_{t-1}$ corresponding to the output of the convolutional GRU $\text{CGRU}_\theta$ from the previous time step. If $t = 0$ then $\widetilde{h}_{t-1}$ are tensors filled with zeros.

- The 512-dimensional hidden states $h^o_{t-1}, h^c_{t-1}$ corresponding to the output of the LSTM $\text{LSTM}_\theta$ from the previous time step. If $t = 0$ then these tensors are identically zero.

- The agents' current metric map of it's environment, this includes a 144-dimensional vector for every unique location tuple the agent has visited.

- (OH only) If the last action taken in the OH-episode was a `PlaceAt`, an action embedding specifying the location location the OM-episode placed the object, e.g. if the OM-episode placed the object in the $C_{ij}$ grid location with modality BEHIND, then the agent obtains the embedding for a successful `PlaceAt|2,i,j` action. As an object can potentially land in many grid locations and modalities simultaneously, the chosen action corresponds to the location and modality containing the largest number of the object's visible pixels.

- (OM only) The modality $m$ and grid location $i, j$ where the object should be placed. This triple is used to slice into from a parameter tensor of size $3 \times 8 \times 13 \times 13$ retrieving a $8 \times 7 \times 7$ tensor which is concatenated to the $128 \times 7 \times 7$ tensor of visual features.

Excluding the recurrent states and the map, each of the above embeddings are repeated (if necessary) so that they are of shape $N \times 7 \times 7$ and concatenated together with the $128 \times 7 \times 7$ SIR. For instance, the 16-dimensional object embedding becomes a $16 \times 7 \times 7$ tensor $T$ where $T_{:,i,j}$ is a copy of the original embedding for each $i, j$. To this tensor is additionally concatenated the first 256-dimension of $h^o_{t-1}$ after having been tiled to be of

shape $256 \times 7 \times 7$. This concatenated tensor is then fed into $\text{CGRU}_\theta$ along with the hidden state $\widetilde{h}_{t-1}$ producing $\widetilde{h}_t$.

The tensor $\widetilde{h}_{t-1}$ is then split into two components. It's final channel, $\widetilde{h}_{t-1}[256, :, :]$, is removed and becomes the tensor $W^{s,t}$ described in Sec. C.4. The remaining $256 \times 7 \times 7$ tensor is then averaged pooled to create the 256-dimensional vector $\overline{h}_t$.

Now the network reads from the input metric map using both an attention mechanism and a collection of convolutional layers resulting in an embedding of size 356. This embedding is concatenated to $\overline{h}_t$ and fed, along with the hidden states $(h^o_{t-1}, h^c_{t-1})$, into $\text{LSTM}_\theta$ producing the hidden states $(h^o_t, h^c_t)$.

For each stage $X \in \{\text{E\&M, OH, OM, S}\}$ there is now a unique MLP $\text{MLP}^X_\theta$ corresponding to a $512 \times 512$ linear layer followed by a ReLU non-linearity. Additionally there are unique networks $\text{ACTOR}^X_\theta$ (corresponding to a $512 \times$ (number of actions in stage $X$) linear layer followed by a softmax nonlinearity) and $\text{VALUE}^X_\theta$ corresponding to a $512 \times 1$ linear layer.

Now the agents' policies and values for each stage are computed as $\text{ACTOR}^X_\theta(\text{MLP}^X_\theta(h^o_t + h^o_t))$ and $\text{VALUE}^X_\theta(\text{MLP}^X_\theta(h^o_t + h^o_t))$ respectively.

Finally the agents then write into their map, at a position defined by their current location tuple, a 128-dimensional vector extracted from one of the final layers of the $\text{CNN}_\theta$ (when applied to $i_t$) along along with the 16-dimensional object embedding.

## D.2 Metric mapping architecture

Intuitively, our metric mapping architecture is designed so that an agent can record information regarding its current state for later retrieval. This map is called *metric* as data is written into the map so that it is spatially coherent, that is, where data is written in memory is uniquely defined by the agent's location tuple when it is written. Our metric map model implements is designed to allow for the following.

- Writing: given the current location tuple of an agent, passes a given value through a Gated Recurrent Unit (GRU) (Cho et al., 2014) whose hidden state is taken from the last value saved at that location (if any, otherwise a learned initial state is used) and this output of the GRU then overwrites the previously saved values.

- Reading at location: given a location tuple, returns the value saved at that location.

- Reading with attention: given an input key, uses a soft-attention mechanism to return a weighted sum of saved values.

- Reading as tensor: return a $4 \times 2 \times 144 \times h \times w$ tensor $M$ where $M_{r,s,:,i,j}$ is the value written into the map when the agent was most recently at location tuple

$$(\min(x \text{ values written into map}) + j \cdot 0.25, \max(z \text{ values written into map}) - i \cdot 0.25, s, r),$$

$h$ is the number of distinct $z$ values visited by the agent, plus optional padding, as it built the map, and $w$ is the number of distinct $x$ values written into the map, again possibly with optional padding. Notice that decreasing $i$ in $M_{r,s,:,i,j}$ results in reading from locations north of the current location and similarly increasing $j$ results in reading locations more east.

- Reading egocentric map: the same as reading as map but with spatial coordinates centered on the agent and appropriately transformed so that north is whichever direction the agent is currently facing.

Using the above, the agent can build a map while it explores its environment while querying that map locally and globally so as to make decisions.

## D.3 Visual dynamics replay decoder architectures

Recall from our training and losses section, that in VDR we are given a triplet $(i_0, a, i_1)$, where $i_0, i_1$ are two images with $i_1$ occurring after an agent took action $a$, and must, accomplish a number of self-supervised tasks. To accomplish these tasks we require several neural network architectures building upon our SIR generating architecture $\text{CNN}_\theta$. These architectures are all convolutional neural networks with the largest of these networks is inspired by spatial transformer networks (Jaderberg et al., 2015), as well as U-Nets, and contains 13 convolutional layers, 10 of which are transposed convolutions which work to upsample the input tensor, the SIR output from $\text{CNN}_\theta$, to a final size of $16 \times 224 \times 224$ after which $1 \times 1$ convolutions are used to produce $\widetilde{i}_1^{224 \times 224}$ as well as $T^{\text{confidence}}$ and $T^{\text{pix. diff.}}$. The tensor $T^{\text{would succeed}}$ is produced by a convolutional network acting on $\text{CNN}_\theta(i_0)$ and the embedding of $a$ while the value $v^{\text{inv. dyn.}}$ is produced by a convolutional network applied to a concatenation of $\text{CNN}_\theta$'s embeddings of both $i_0$ and $i_1$. Implementation details can be found in our code base.

## D.4   PERSPECTIVE SIMULATION ARCHITECTURE

The perspective simulation architecture operates on the tensor representation of the map, see the metric mapping architecture above, which is reshaped to be of size $1152 \times h \times w$ where $h$ and $w$ are the height and width of the map respectively. First this map is concatenated with a grid of distances from the agent start location after which it is processed by several $1 \times 1$ convolutional blocks to form the intermediary representation $M^1$ of the map. An additional $1 \times 1$ convolution produces, for each location tuple $\ell$ (merging those which only differ in that the agent is, or is not, standing) after and after reshaping, two tensors $P^\ell$ and $V^\ell$ of size $2 \times 3 \times 7 \times 7$ as well as two values $p_{none}^\ell, v_{none}^\ell \in \mathcal{R}$. For simplicity we will abuse notation and let $P_{none}^\ell = p_{none}^\ell$ and $V_{none}^\ell = v_{none}^\ell$. Next, given a hypothetical hiding outcome $e \in \mathcal{S}^{\text{OH outcomes}}$ another convolutional neural network, acting on $M^1$ along with an embedding of $e$ indicating where in $M^1$ the object is being hidden, produces the second intermediary representation of the map $M^e$. A "mental" seeker agent is then defined in this architecture using an LSTM whose observations come from slices of $M^e$. This mental seeker can move about the map defined by the representation in $M^e$ and, at each step $i$, produces the vectors $\widetilde{\pi}_i$ and $\widehat{s}_i$ defined in the training in losses section by using two linear layers acting on the mental agent's hidden state.

## E   HUMAN AND SELF EVALUATION FOR CACHE AGENTS

In Figures 4a to 4c we report the results of evaluating our hiding and seeking agents from three points during their training as they compete against one another as well as against hiding places chosen by an exhaustive approach and hiding places chosen by humans. In the following we discuss how we collected hiding locations for the various agents and ran our seeking trials.

### E.1   GENERATING HIDING LOCATIONS

**Human agents.** We recruited 16 human volunteers to hide objects within AI2-THOR. Participants were instructed that they would be randomly positioned in a simulated environment holding an object which they were then expected to hide so that a seeking agent, who started in the same position that they started in, would find the object difficult to find. Furthermore participants were instructed to be in the mindset that they might be playing against the same seeker many times and thus should choose diverse hiding locations.

Before beginning, participants were allowed to familiarize themselves with the interface until they judged themselves sufficiently competent to begin the task. Unlike our artificial agents, humans subjects were not constrained in the number of actions they were allowed to take and, after hiding their object, were allowed to pick it up again and try other hiding locations if desired.

We collected 600 human hiding locations evenly distributed among all object types and training, validation, and test scenes. To reduce the amount of data to be collected we used only ten (of 40) training scenes (AI2-THOR floor plans 7, 10, 13, 16, 18, 203, 209, 213, 215, and 218).

**Artificial agents.** We generate hiding locations from our artificial hiding agent using the agent after it had trained for 0, $10^6$ and $4.5 \cdot 10^6$ episodes. We call these agents $h_0$, $h_{1e6}$, and $h_{4.5e6}$. For each of these agents we generate 4,000 total hiding locations, 2,000 each for the train and test scenes. The train (test) hiding locations are generated such that for each (scene, object type) pair we have an equal number of hiding locations. For fair comparison, we also generated a set of 1,545 additional hiding locations from $h_{4.5e6}$ using the same scenes, object types, and agent starting locations as those from the human trials (with multiple repeats per agent start location).

**Exhaustive approach.** Recall the exhaustive automated approach for generating hiding places of easy, medium, and hard difficulties detailed in the cache stages section. Using this approach we generate, for each of these difficulties, 3,999 hiding locations in the train scenes and 998 in the test scenes. Because of our automated, metric-based, approach for generating these hiding spots there were three cases (one in the train scenes and two in the test scenes) where there were not be sufficiently many hiding locations 20 desired locations per object, scene, and difficulty. In such cases we reduced the number of locations saved for that object and scene pair for each difficulty level so that the total number of hiding places in each difficulty type remained equal.

### E.2   RUNNING SEEKING TRIALS

In these trials a seeking agent is required to find an object hidden in a hiding location from one of the above collections. If, through automated checks, we find that a hiding place is such that the hidden object cannot be seen from any reachable position in the scene then this hiding location is never shown to any seeking agent and all seeking agents are considered to have failed to find the object for that trial.

**Artificial agents.** We run seeking trials using the artificial seeking agents from the same three points in training and name these seeking agents analogously as $s_0$, $s_{1e6}$, and $s_{4.5e6}$.

**Human agents.** For seeking we recruited human participants using Amazon's Mechanical Turk platform. As in the hiding task participants were allowed to familiarize themselves with the interface until they judged themselves sufficiently competent to begin the task and were not constrained in the number actions they were allowed to take.

On starting a trial, the participant was randomly assigned a hiding location generated either by $h_{4.5e6}$ or from the set of 600 human locations. To ensure fair comparison, we matched each of the 600 human locations to a unique hiding location generated by $h_{4.5e6}$ (from the additional set of 1,545) where the artificial hiding agent started the same position as the human hider. After 100 seconds participants were allowed to give up, moving them to the next seeking trial, if they felt the object could not be found.

At the end of these trials we were left with complete seeking trials for all but two instances where invalid data was returned.

## F    STATIC IMAGE REPRESENTATION EXPERIMENTS

In these experiments we consider SIRs produced by the model $\text{CNN}_\theta$ when trained:

(1)  to play cache (taken our final cache agent),

(2)  to do joint object class prediction using precisely the same images seen by our cache agent during training (here an object is considered present in an image if AI2-THOR returns that the object is visible)

(3)  to complete a navigation task (see Sec. G.1),

(4)  using an autoencoding loss (i.e. pixel-wise mean absolute error) with 35,855 AI2-THOR images, or

(5)  with an image classification loss on the 1.28 million hand-labeled images in ImageNet (Krizhevsky et al., 2012).

More details regarding these baselines can be found in Sec. . We begin by describing our experiments for synthetic images before moving to our natural image experiments.

### F.1    SYNTHETIC IMAGES

#### F.1.1    EVALUATION DATA AND TASKS

To evaluate the quality of our SIRs we train decoder models on these SIRs to complete a number of distinct tasks described below. Except for in the classification task, where the decoder model is a max-pooling layer followed by a linear layer, the decoder models are designed similarly to those from a U-Net (Ronneberger et al., 2015), for details please see our code.

**Depth prediction.** In AI2-THOR we generate a collection of $3 \times 224 \times 224$ RGB images paired with their corresponding $224 \times 224$ depth maps. Models are trained to predict the depth map given its corresponding RGB image by minimizing a per-pixel mean absolute error loss.

**Surface normals.** In AI2-THOR we generate a collection of $3 \times 224 \times 224$ RGB images where each image is paired with a corresponding $3 \times 224 \times 224$ tensor $T$ where $T_{:,i,j}$ represents the surface normal at pixel $i, j$ in the RGB image. Models are trained to predict this tensor of surface normals given its corresponding RGB image by minimizing a per-pixel negative cosine similarity loss.

**Classification.** In order to generate a dataset for classification we first select 90 types of objects available in AI2-THOR and assign each of these types a unique index in 1, ..., 90. Next, in AI2-THOR, we generate a collection of $3 \times 224 \times 224$ RGB images where each image is paired with a $\{0, 1\}$-valued vector of length 90 whose $j$th entry equals 1 if and only if an object whose type has index $j$ is present in the image and whose pixels occupy at least 1% of the image. Models are trained to predict this vector of 0,1 values from the input image using binary cross entropy.

**Openability.** In AI2-THOR we generate a collection of $3 \times 224 \times 224$ RGB images where each image is paired with a a $224 \times 224$ $\{0, 1, 2\}$-valued tensor $T$ with entry $T_{i,j}$ equaling 1 if pixel $i, j$ in the input image corresponds to an object that can be opened, equalling 2 if the pixel corresponds to an object that can be closed, and otherwise equals 0. Models are trained to predict the $\{0, 1, 2\}$-valued tensor from the corresponding RGB image by minimizing a per-pixel mean cross entropy loss where the loss associated with predicting a 0 has a weight of 0.2 and the other losses have a weight of 1.0.

**Traversable surfaces prediction.** In AI2-THOR we generate a collection of $3 \times 224 \times 224$ RGB images where each image is paired with a corresponding $224 \times 224$ $\{0, 1\}$-valued tensor $T$ with $T_{i,j}$ equalling one if and

only if the $ij$th pixel in the image corresponds to a surface onto which the agent can walk. Models are trained to predict the $\{0, 1\}$-valued tensor given its corresponding RGB image by minimizing a per-pixel binary cross entropy loss.

**Object depth.** In AI2-THOR we generate a collection of $3 \times 224 \times 224$ RGB images each where each image is associated with a collection of $224 \times 224$ $\{0, 1\}$-valued masks of, sufficiently large, objects in the image paired with depth maps corresponding to the image in which the masked object has been removed. Models are trained to predict, from an input RGB image and object mask, the depth map of the image where the object has been removed by minimizing a per-pixel weighted mean absolute error loss. In this loss, pixels corresponding to the masked object are weighted to contribute equally to those pixels where the object is not present.

**Object surface normals.** Similar to object depth but predicting surface normals after an object has been removed.

### F.1.2 TRAINING

We train using an ADAM optimizer with $(\beta_1, \beta_2) = (0.99, 0.999)$, a weight decay of 2e-6, and a plateau-based learning rate scheduler beginning with a learning rate of 1e-2 and decaying by a factor of 10 with a patience of $N^{\text{patience}}$, that is, the scheduler reduces when the loss on the validation set has failed to decrease by some small absolute quantity for $N^{\text{patience}}$ subsequent epochs). When training $\text{CNN}_\theta$ to perform auto-encoding and ImageNet classification we used a patience of $N^{\text{patience}} = 5$ and trained the models using all available training data. When training our other tasks, we used a patience of $N^{\text{patience}} = 5$ and subsampled the training data to include 1,200 examples. Training was stopped if the learning rate ever decreased below 1e-4. During training models were evaluated on the validation set after every epoch and, once training completed, the model with the best validation loss was returned as the final model.

### F.1.3 EVALUATION

To evaluate our models we compute the loss of each model on each of the test set examples for its respective task. As any test set results generated within the same scene and at nearby location tuples are correlated, we use a generalized linear mixed effects model (GLMM) to obtain statistically meaningful comparisons between our models. We run our analysis using the R (R Core Team, 2019) programming language, in particular we use the `glmmPQL` function in the `nlme` (Pinheiro et al., 2019) package to fit our GLMM models and then the `emmeans` (Lenth, 2019) package to obtain p-values of contrasts between fixed effects. In these analyses we:

- Let the per test set example loss be our endogenous variable.
- Estimate fixed effects corresponding to which SIR was used to produce the loss (auto-encoder, ImageNet, or cache).
- Include mixed effects corresponding to scene nested within scene type.
- Model spatial correlation, nested within scene, using spherical correlation where two test set examples corresponding to different SIRs or with different rotations are considered infinitely distant.
- Use a Gamma family model with an identity link.
- Subsample the test set to include at most 50 datapoints per scene and SIR so that the model can be more quickly trained.

To fully reproduce our results please see our code base.

## F.2 NATURAL IMAGES

We consider three distinct tasks for natural images, scene classification on the SUNCG dataset Xiao et al. (2010), depth estimation on the NYU V2 dataset Nathan Silberman & Fergus (2012), and walkable surface estimation also on the NYU dataset. As for the synthetic image experiments, we use $\text{CNN}_\theta$ as our backbone model.

### F.2.1 EVALUATION DATA AND TASKS

**Scene Classification.** In scene classification the goal is to predict, from a natural image, the type of scene the image is staged in. For instance, scene categories include beach, palace, and sauna. We evaluate our predictions using the (mean-per-class) top-1 accuracy metric.

**Depth prediction.** As for synthetic images, our goal is to predict pixel-to-pixel depth but for natural images. We evaluate our model using the standard absolute relative error of our predictions.

**Walkable surface estimation.** This is synonymous with traversable surfaces prediction in AI2-THOR. We evaluate the quality of our predictions using the standard IOU (intersection over union) metric where we only consider the IOU with walkable segments.

### F.2.2 Training

**Scene Classification.** Our decoder architecture for scene classification corresponds to a 1-by-1 convolutional layer which transforms the $128{\times}7{\times}7$ SIR output from $\text{CNN}_\theta$ to a $64{\times}7{\times}7$ tensor. Two fully connected layers then transform this tensor into a vector of size 512 followed by vector of size 397. For training we use the cross-entropy loss.

**Depth estimation.** To perform depth prediction our network $\text{CNN}_\theta$ is composed with a feature pyramid network (Lin et al., 2016) and pixel shuffle layers Shi et al. (2016) are used for upscaling. The Huber loss is minimized for training.

**Walkable surface estimation.** This architecture is essentially identical as for depth prediction. The architecture of this network is the same as the depth estimation network. Our training object is pixel-wise binary cross entropy.

## G Dynamic image representation experiments

In the following we describe our training procedure for our dynamic image experiments along with any task-specific details including definitions of our baselines, namely the labels in the legend of Fig. 7. For tractability of inference, we will always consider our training set $D_{\text{train}}$ to be fixed and ignore any randomness, if applicable, in the generation of $D_{\text{train}}$. Each of our considered tasks is a discrimination task and requires predicting, from a given representation, whether or not some binary condition holds. We call cases where the condition holds *positive examples* and others *negative examples*. We will accomplish these predictions by fitting a logistic regression models, details are given in the training section below.

### G.1 Baselines

We will now describe our baselines referenced in the labels of Fig. 7 and in Sec. 5.

**DIR (Cache).** The is the DIR generated by our trained cache agent. In the tracking experiment (the experiment corresponding to Fig. 7b) this DIR corresponds to the $257{\times}7{\times}7$ tensor $\widetilde{h}_t$ (recall the notation from Sec. 4) produced by the trained $\text{CGRU}_\theta$ model. In all other experiments this corresponds to the 512-dimensional vector $h_t^o$ produced by the $\text{LSTM}_\theta$ architecture.

**DIR (Nav.).** For comparison we trained a *navigation agent* to perform the seeking task with object locations chosen from the sets of hiding locations, of easy and medium difficulty, generated by our exhaustive procedure. This navigation agent was trained, without VDR, for 150,000 episodes at which point performance appeared to saturate. The navigation agent was trained with a similar architecture to our cache agent with a convolution GRU but without a final $\text{LSTM}_\theta$ model. For the tracking experiment the DIR for this agent was taken as the $257{\times}7{\times}7$ tensor output from its convolutional GRU. For the other experiments, the DIR was taken as a weighted average pool of the aforementioned $257{\times}7{\times}7$ tensor. Note that, unlike the cache agent, the navigation does not have any actions corresponding to moving the object in its hand. Because of this, in the below experiments, whenever we would tell the navigation agent that its last action was a hand movement action we instead tell it that its last action was a `Stand` action.

**DIR (Rand.)** It is well-known that convolutional networks structurally encode powerful inductive biases for image processing and that even randomly initialized networks can provide surprisingly powerful features for downstream tasks. This raises the question of whether or not the cache DIR performs well simply because our model architecture encodes strong inductive biases and, perhaps, that we do not need to train our model at all to obtain good results for our DIR tasks. To address this question, DIR (Rand.) is generated identically to DIR (Cache) except that the underlying model weights are kept as their randomly initialized values.

**SIR (Cache).** This baseline corresponds to the features produced by $\text{CNN}_\theta$ trained by playing cache. For the tracking experiment we use the entire $128{\times}7{\times}7$ SIR produced by this CNN, in all other experiments we average pool this tensor to reduce it to a 128-dimensional vector.

**SIR (ImageNet).** This model is identical to SIR (Cache) except the underlying CNN was trained with an image classification loss on the 1.28 million hand-labeled images in ImageNet (Krizhevsky et al., 2012).

**SIR (Class.)** One may wonder if a CNN trained to perform classification on AI2-THOR images would obtain similar performance to our cache-trained model. To answer this question, while training our cache agent we have, in parallel, trained a $\text{CNN}_\theta$ model to classify AI2-THOR objects. Critically, this model was trained with

exactly the same set of images as our cache agent and received exactly the same number of gradient updates. Objects were considered to be present in an image if they were considered visible by AI2-THOR. To train this model we minimized a binary cross entropy loss after average pooling the final representation from $\text{CNN}_\theta$ to obtain a 128-dimensional vector after which we applied a $128{\times}101$ linear transformation giving us the 101 binary logits used for prediction (of the 101 applicable AI2-THOR objects). As for SIR (Cache), we use the $128{\times}7{\times}7$ tensor for the tracking experiment and, for all other experiments, average pool this vector to be of size 128.

**Forward Steps.** This baseline is only included in the "Contained or Behind?" experiment. In this experiment there is a non-trivial bias in the dataset, namely that an object needs to be moved further forward to be placed behind a receptacle than it does to be placed into it. To ensure that DIR (Cache) is not simply exploiting this bias, we have included the count of the number of forward steps as a baseline.

**DIR (4CNN-Cache/Class/ImgNet).** An straightforward strategy by which to produce a DIR from SIRs is concatenation. Namely, if we have an SIR $s_t$, then it is we may build a representation that "integrates observations through time" simply by concatenating several such SIRs together. That is, a DIR representation at time $t$ could simply be the concatenation $[s_{t-3}, s_{t-2}, s_{t-1}, s_t]$. The three baselines DIR (4CNN-Cache), DIR (4CNN-Class), and DIR (4CNN-ImgNet) each take this approach by concatenating the SIR from time $t$ with the SIRs from the three previous timesteps. For DIR (4CNN-Cache), SIR (Cache) representations are concatenated (and similarly for DIR (4CNN-Class), and DIR (4CNN-ImgNet)). For the tracking experiments these DIRs are thus of shape $512{\times}7{\times}7$ while for the the other experiments these DIRs are vectors of length 512. As we wish these DIRs to be strong baselines, in the "Contained or Behind?" experiment we also concatenate to these DIRs the Forward Steps feature described above (producing 513-dimensional DIRs).

## G.2    TRAINING

We wish to evaluate how our considered DI- and SI- representations perform in our tasks as the amount of data available for training varies when using a training procedure that follows best practices in machine learning by using dimensionality reduction and cross validation. Our procedure, based on iteratively subsampling $D_{\text{train}}$, follows. Suppose we are interested in our representations' performance given a training set of size $n > 0$ with $n < |D_{\text{train}}|$. We then repeat the following $R > 0$ times.

1. Subsample a training set $D_n \subset D_{\text{train}}$ of size $n$ taking care to retain as close to the same proportion of negative and positive examples as possible.

2. Perform dimensionality reduction on $D_n$ using (a variant of) principal component analysis, reducing the datasets dimensionality so that at least 99.99% of variance is explained by the remaining components, to obtain a dataset $\widetilde{D}_n$.

3. Fit a logistic regression model on $\widetilde{D}_n$ using cross validation to choose an appropriate weighting of $\ell_2$-regularization.

4. Apply the principal component analysis transformation learned in the second step to the test set $D_{\text{test}}$ to obtain a $\widetilde{D}_{\text{test}}$ and then compute the test-accuracy on this transformed dataset.

This gives us $R$ accuracy values $a_{n1}, ..., a_{nR}$ from which we can estimate the accuracy of our model on test examples when training our model on a randomly sampled training set of size $n$ as the mean $\overline{a}_n = \frac{1}{R}\sum_{i=1}^{R} a_{ni}$. Note that randomness in the estimator $\overline{a}_n$ comes both from our resampling procedure and, assuming our test set is not exhaustive, from the randomness in which examples where chosen to be included in the test set. The first source of randomness can be reduced simply by increasing $R$ while the second can only be reduced by gather more test set examples. When appropriate, we account for both sources of randomness when plotting confidence intervals.

## G.3    CONTAINED OR BEHIND RELATIONSHIPS

For these experiments we generate a dataset using the first five scenes of the foyer type in AI2-THOR. For each of these five scenes we generate a large collection of paired examples of objects being places into, or behind, one five receptacle objects while that receptacle sits on one of five possible tables at varying offsets from the agent. The placed objects are the same as those hidden in cache and we use five different receptacle objects of varying size. In order to make our setting especially difficult, we modify the above training procedure so that, in step 1 above, we restrict our subsampled training set to exclude a randomly selected combination of scenes, objects, receptacles, and tables. In step 4, our testing set is those examples in which the scene, object, receptacle, and table are all among those excluded from the training set. This makes the test examples especially challenging as they are, visually, very dissimilar from those see during training. In this setting, only consider randomness as a result of our resampling procedure in our error estimates and so our inference is restricted to within our generated set of examples.

Here predictions are made using the representation at the timestep after the object has been moved into its position but before it has been let go by the agent (*i.e.* before a `DropObject` action has been taken).

## G.4  OCCLUDED OBJECT TRACKING

As in our contained or behind relationship experiments we generate a dataset using the first five scenes of the foyer type in AI2-THOR and five table types. Here, as we are interested in tracking completely occluded objects, we use only three of our five item items, the cup, knife, and tomato, as these are sufficiently small to be frequently occluded. For each configuration of object, table, and scene, we generate a collection of object trajectories as the object moves behind a collection of books, recall Fig. 7b. For additional variability we produce many such trajectories with the books positioned in different distances and orientations. As the object moves behind the occluding books, we record all grid locations $C_{ij}$ that the object passes through into the set $P$ (it is considered to have passed through a grid location $C_{ij}$ if, at any point during its trajectory behind the books, at least 10 pixels of the object were in $C_{ij}$ after making all other objects invisible). At every step of a trajectory where the object is occluded such that less than 10 pixels of the image correspond to the object we generate $|P|$ data points, one for each grid location $C_{ij} \in P$ with the grid location being considered a positive example if, after making all other objects invisible, the moving object has greater than 10 pixels in that grid location and otherwise being consider negative. In the paragraph below we describe how we obtain the SIR and DIR features corresponding to each such data point. Note that, as objects tend to occupy only a small portion of an image, there are many more negative examples in our dataset than positive ones. During training we always subsample negative examples so that there is an, approximately, equal number of positives and negatives. Furthermore, during testing, we implicitly balance positive negative examples by reweighting; thus the accuracy we report in this setting is a *balanced* accuracy. As in the previous experimental design, we modify the above training procedure so that, in step 1 above, we restrict our subsampled training set to exclude a randomly selected combination of scenes, objects, and tables. In step 4, our testing set is those examples in which the scene, object, and table are all among those excluded from training. Again as above, only consider randomness as a result of our resampling procedure in our error estimates and so our inference is restricted to within our generated set of examples.

Recall from our Sec. G.1, all SIRs and DIRs in this experiment have shape $N \times 7 \times 7$ for some $N \geq 0$. Let $T$ be such an SIR or DIR and suppose that, at time $t > 0$, the object is fully occluded and $X$ is a $7 \times 7$ $\{0, 1\}$-valued matrix with $X_{ij}$ equalling one only if grid position $C_{ij}$ is considered a positive example. To obtain, from $T$, the features used to predict $X$ we simply take a slice from $T$ at position $i, j$. Namely the features used to predict the value of $X_{ij}$ are $T[:, i, j]$.

## G.5  OBJECT PERMANENCE WHEN ROTATING

For each of the hiding locations in our automatically generated easy hiding places, recall Sec. E.1, we let our seeking agent attempt to find the object for 500 steps. If, during one of these episodes, the agent would successfully take a `ClaimVisible` action at time $t$, so that it has found the object, we consider the counterfactual setting in which the agent instead rotates right or left once at random and the, previously seen, object vanishes. The agent then is allowed to take 4 more actions following its policy. Each of the five representation generated after the rotation is saved as a positive example. To create negative examples we replay this same episode, forcing the seeking agent to take all of the same actions, but with the goal object invisible during the entire trajectory. From this replayed trajectory we save the final five resulting representations as negative examples. When sampling training sets we consider only training scenes and randomly exclude one object, when sampling test sets only testing scenes and examples including the previously excluded object.

## G.6  FREE SPACE SERIATION

In our final experiment we test the ability of our DIRs and SIRs to predict free space. We construct our dataset for this task by, for every AI2-THOR living room and kitchen scene $s$, performing the following procedure.

1. Let $N^s = \min(20, \text{round}(0.15 \cdot \#\text{unique reachable } (x, z) \text{ positions in the scene on a 0.25m grid}))$, and randomly select $(x_1, z_1), ..., (x_{N^s}, z_{N^s})$ locations in $s$.

2. For every position $(x_i, z_i)$ perform the following.

   a. Teleport the agent to $(x_i, z_i)$ with a random (axis-aligned) rotation, and randomly select a direction in which to rotate (left or right). The first set of rotations is a habituation phase.

   b. Suppose we have chosen to rotate right. Then require the agent to take seven `RotateRight` actions so that every orientation is seen exactly twice.

   c. Save the DIRs, and SIRs, corresponding to the agents evaluation when looking in each orientation for the second time along with whether or not the current orientation has strictly more free space than the previous orientation. Free space is measured by the count of the grid locations that the agent could

---

**Algorithm 1:** Exploration and mapping episode reward structure

---

**Input:** Time step $t \geq 0$, full history of E&M episode $H$, number of steps taken in episode $N$
**Output:** Reward for E&M episode step at time $t$.

1 **begin**
2    **for** $i \leftarrow 1, \ldots, N$ **do**
3       $a_i \leftarrow i$th action taken in episode
4       $\text{success}_i \leftarrow$ was $i$th action successful
5       $\text{extrap}_t \leftarrow \mathcal{S}_{E\&M,t}^{\text{extrap.}}$
6       $\text{new\_location}_t \leftarrow$ true if the location of the agent at time $t$ did not occur previously in the episode.
7    $r \leftarrow -0.01$
8    **if** *not success$_t$* **then**
9       $r \leftarrow r - 0.02$
10    **else**
11       `// encourage the agent to explore new areas`
12       `num_new_explored` $\leftarrow$
13       $r \leftarrow r + 0.2 \cdot (\text{extrap}_t - \text{extrap}_{t+1})/3$
14       `// encourage the agent to attempt opening objects`
15       `new_opened` $\leftarrow a_t$ opened an object that was not previously opened in the episode
16       **if** *new_opened* **then**
17          $r \leftarrow r + 0.4$
18       `// encourage the agent to explore areas considered good for hiding`
19       **if** *new_opened or new_location$_t$* **then**
20          $r \leftarrow r + r_{\text{E\&M,hide},t}$
21    **if** $a_t$ *is an OpenAt action* **then**
22       `// never punish the agent heavily when taking an OpenAt action`
23       $r \leftarrow \max(r, -0.001)$
24    **return** $r$

---

potentially occupy within the agent's visibility cone. Cases where a given orientation has strictly more free space are considered positive examples, the others negative.

3. Combine all positive and negative examples into our training and test datasets.

### G.6.1 WHY DO OTHER DIRS OBTAIN HIGH PERFORMANCE?

In Fig. 7d we see that several 4CNN-DIRs do nearly as well as DIR (Cache). Initially this may seem surprising, our other experiments seem to suggest that our cache agent's DIR is significantly more capable than a simple concatenation of SIR features. We argue, however, that this result is not at all surprising. If an SIR can be used to predict the amount of free space in an image to a reasonable degree of accuracy then this task can be tackled using a 4CNN-DIR by simply predicting the free space using the SIRs in the last two frames and then taking their difference. That is to say, this seriation task well suited for 4CNN-DIRs when their underlying SIRs record free space.

---

**Algorithm 2:** Object hiding episode reward structure

---

**Input:** Time step $t \geq 0$, full history of OH episode $H$, number of steps taken in episode $N$
**Output:** Reward for OH episode step at time $t$.

1 **begin**
2    $r \leftarrow -0.01$
3    **for** $i \leftarrow 1, \ldots, N$ **do**
4      $a_i \leftarrow i$th action taken in episode
5      success$_i \leftarrow$ was $i$th action successful
6    **if** $a_t = $ READYFORSEEKER **then**
7      **if** *not* success$_i$ **then**
8        $r \leftarrow r - 0.1$
9    **else if** success$_t$ *and* $a_t$ *is an* `OpenAt` *action* **then**
10      `// encourage the agent to attempt opening objects`
11      **if** *the opened object was not previously opened in the episode* **then**
12        $r \leftarrow r + 0.2$
13    **else if** $a_t$ *is a* `PlaceAt` *action but a* `PlaceAt` *action was previously successful* **then**
14      $r \leftarrow r - 0.1$
15    **else if** success$_t$ *and* $a_t$ *is a* `PlaceAt` *action* **then**
16      `// Reward successful manipulation and encourage the agent`
         `to try hiding spots that are more difficult to manipulate`
         `to`
17      $p \leftarrow$ the probability the object manipulation module assigns to successfully accomplishing $a_t$
18      $r \leftarrow r + 0.02 + 1/2 \cdot \min(\max(p, 0.0001)^{-2}), 100)/100$
19    **else if** *not* success$_t$ **then**
20      $r \leftarrow r - 0.01$
21    **if** *there was a successful* `PlaceAt` *action taken in the episode and* $a_t$ *is the last successful action that is not a* READYFORSEEKER *action* **then**
22      $r \leftarrow r + 5 \cdot r_{OH,\text{percentile}}$
23    **return** $r$

---

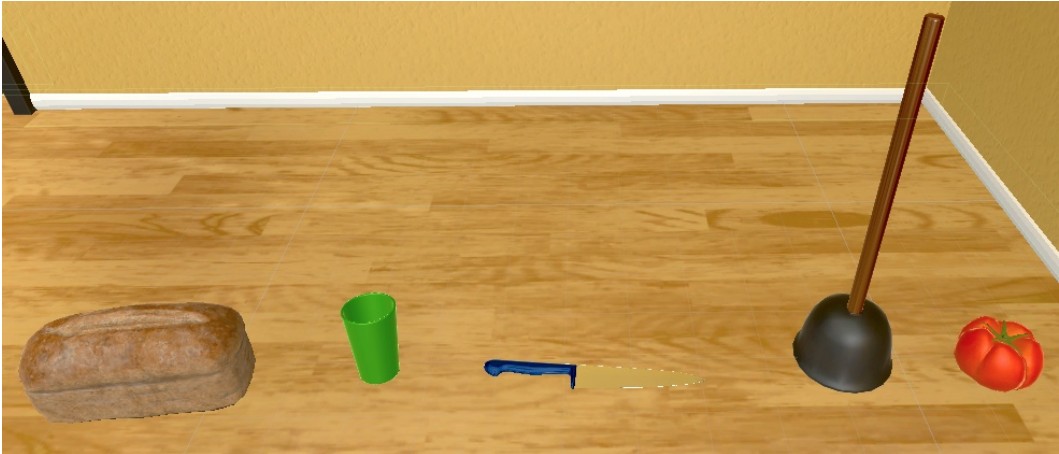

Figure G.1: The five objects hidden that can be hidden during a cache game. From left to right these are a loaf of bread, cup, knife, plunger, and tomato.

---

**Algorithm 3:** Object manipulation episode reward structure

---

**Input:** Time step $t \geq 0$, full history of OM episode $H$, number of steps taken in episode $N$, the target location specified by the action `PlaceAt|m,i,j` recall Table G.1

**Output:** Reward for OM episode step at time $t$.

1 **begin**
2     `hand_locations` = empty list
3     **for** $k \leftarrow 1, \ldots, N$ **do**
4        $a_k \leftarrow k$th action taken in episode
5        $\text{success}_k \leftarrow$ was $k$th action successful
6        $(x, y, z) \leftarrow$ world coordinates of agent's hand at time $k$ rounded to 2 dec. places
7        Append $(x, y, z)$ to the end of `hand_locations`
8     $r \leftarrow -0.01$
9     **if** *not success$_t$* **then**
10        $r \leftarrow r - 0.02$
11     **else if** $a_t$ *is an* `OpenAt` *action* **then**
12        `// Encourage the agent to attempt opening objects`
13        **if** *the opened object was not previously opened in the episode* **then**
14           $r \leftarrow r + 0.1$
15     **else if** $a_t = $ `DropObject` **then**
16        `n_obj_pixels` $\leftarrow$ total number of pixels corresponding to the object after making all other objects invisible
17        **if** `n_obj_pixels` $\leq 10$ **then**
18           `// Discourage the agent from placing objects off screen`
19           $r \leftarrow r - 1.0$
20        **else**
21           `hit` $\leftarrow 3 \times 7 \times 7$ tensor recording the number of pixels of the object dropped object in each of the grid locations and modalities, recall the discussion in Sec. B.4 and Fig. 3d
22           $M \leftarrow$ Compute $M$ as in described in Sec. B.4
23           `hit_inds` $\leftarrow$ list of the multi-indices of `hit` where the corresponding entry of `hit` is $\geq 0.95 \cdot M$
24           `// Encourage the agent to place the object close to the goal in pixel space (ignoring modality)`
25           $r \leftarrow r + 0.25^{\min\{|i-b|+|j-c| \, : \, (a,b,c) \in \text{hit\_inds}\}}$
26           `// Encourage the agent to place the object in the correct modality (only for` BEHIND `and` CONTAINEDIN `modalities)`
27           **if** $m \neq 0$ *and* $a = m$ *for any* $(a, b, c) \in$ `hit_inds` **then**
28              $r \leftarrow r + 1$
29           `// Give extra reward if the agent got everything exactly right`
30           **if** $(m, i, j)$ *is in* `hit_inds` **then**
31              $r \leftarrow r + 1$
32     **else if** $t = N$ **then**
33        $r \leftarrow r - 1.0$
34     **return** $r$

---

---

**Algorithm 4:** Seeking episode reward structure

---

**Input:** Time step $t \geq 0$, full history of OM episode $H$, number of steps taken in episode $N$, the goal object `goal`

**Output:** Reward for S episode step at time $t$.

1 **begin**
2    **for** $i \leftarrow 1, \ldots, N$ **do**
3       $a_i \leftarrow i$th action taken in episode
4       $\text{success}_i \leftarrow$ was $i$th action successful
5       $\texttt{new\_location}_t \leftarrow$ action $a_t$ resulted in the agent entering a previously unvisited location tuple when ignoring rotation
6       $\texttt{new\_opened}t \leftarrow$ action $a_t$ resulted in the agent opening a previously unopened object
7    $r \leftarrow -0.01$
8    **if** *not success$_t$* **then**
9       **if** $a_t = \text{CLAIMVISIBLE}$ **then**
10          `// penalize the agent for claiming the object is visible when it cannot be seen`
11          **if** *the action failed as the object was not visible in the viewport and not because the agent was too far away from the visible object* **then**
12             $r \leftarrow r - 0.05$
13       **else**
14          $r \leftarrow r - 0.02$
15    **else if** $\texttt{new\_location}_t$ **then**
16       `// don't use a step penalty if the seeking agent entered a new location`
17       $r \leftarrow r + 0.01$
18    **else if** $\texttt{new\_opened}_t$ **then**
19       $r \leftarrow r + 0.06$
20    **else if** $a_t = \text{CLAIMVISIBLE}$ **then**
21       `// the agent has successfully found the object` $r \leftarrow r + 1.0$
22    **return** $r$

---

| Action | Success condition | Impact upon success | Ep. |
|---|---|---|---|
| `MoveAhead`, `MoveLeft`, `MoveRight` | No obstructions 0.25m in the direction of movement. | Agent moves 0.25m forward, left, or right respectively. | E&M, S |
| `RotateLeft`, `RotateRight` | If the agent is holding an object, it will not collide after rotation. | The agent rotates $90°$ in the specified direction. | E&M, S |
| `Stand`, `Crouch` | Agent must begin crouching if taking a `Stand` action and must begin standing if taking a `Crouch` action. Additionally, if the agent is holding an object, this object will not collide with anything when the agent changes height. | Agent stands or crouches. | E&M, OH, S |
| `MoveHandAhead`, `MoveHandLeft`, `MoveHandRight`, `MoveHandBack`, `MoveHandUp`, `MoveHandDown` | Agent must be holding an object, if so the agent applies a constant force on the object in the desired direction until it reaches a max velocity of 1m/s. The agent stops the object if it either moves a total of 0.1m from its starting position, the object reaches a max distance (1.5m) from the agent, the object exits the agent's viewport, or four seconds have elapsed. Once the agent stops the object, if the object has not moved more 0.001m or rotated at least $0.001°$ then the movement is considered failed and the object is reset to its starting position. Otherwise hand movement was successful. | The hand object remains in its new position. | OM |
| `RotateHand\|D`, $D \in \{$`-X`, `+X`, `-Y`, `+Y`, `-Z`, `+Z`$\}$ | Agent must be holding an object, if so the agent rotates the object by 30 degrees along the axis specified by `D`. So if `D` $=$ `-Y` then the object is rotated 30 degrees counter-clockwise along the y-axis. If this rotation would result in the object colliding with another object then the action fails and the object is reset to its starting rotation. | The hand object remains in its rotated orientation. | OM |
| `DropObject` | The agent is holding an object. | The object is dropped. | OM |
| `OpenAt\|i,j`, $1 \le i, j \le 7$ | Recall the partitioning of an image into a 7×7 grid as shown in Fig. 1d. There is an openable object near the center of the $i, j$th grid location. | The object near the location specified by $i, j$ opens. | E&M, OH, OM, S |
| `CloseObjects` | There is at least one object within the agent's interactable range which is open. | All open interactable object are closed. | E&M, OH, S |
| `PlaceAt\|m,i,j`, $1 \le i, j \le 7$, $m \in \{0, 1, 2\}$ | See the description of the object hiding episode. Here the 0,1,2 values for $m$ correspond to the modalities ONTOP, CONTAINEDIN, and BEHIND respectively. | An object manipulation episode begins with target specified by $i, j, m$. | OH |
| `ReadyForSeeker` | The agent has successfully performed a `PlaceAt` action. | The hiding episode ends and the seeking episode begins. | OH |
| `ClaimVisible` | The goal object is within 1.5m of the agent and at least 10 pixels the agents current view can be attributed to the object. | The seeking episode ends successfully. | S |

Table G.1: A description of the actions available to our agents along their success criteria, impact on the environment, and episodes in which they are available.

