# OpenReview forum: "Learning Generalizable Visual Representations via Interactive Gameplay"
_ICLR.cc/2021/Conference — ICLR 2021 Oral_

### Official Review · AnonReviewer3 · 2020-10-20
**Very dense paper, interesting research direction**

**Rating:** 8
**Confidence:** 3

**Review:**

This paper proposes embodied game-playing with artificial agents as a method to learn better representations of their environment. They describe a game, cache, which is a variant of hide-and-seek played in a virtual environment and a method for training an agent to play the game. They present results which demonstrate that the static representations learned through game-playing perform better than other pre-training tasks within the same virtual environment, on both virtual vision and real world vision applications. They also show that the dynamic representations are useful for completing object permanence tests inspired by developmental psychology research.

This research direction (both biologically-inspired CV as a whole and specifically game playing as a pre-training task) is exciting and seems extremely promising. Assuming I understand the methods and results correctly, this seems like a clear accept. My only reservations are the complexity of the task and how the results are presented in the paper, which are relatively minor issues.

The task, Cache, seems extremely complicated to implement and train, since it involves five different stages of embodied exploration/action and adversarial reinforcement learning. While all of these steps are necessary to play the game like humans (or ravens) do, it is unclear to me whether the important learning is occurring as a result of the whole process, or of a specific stage. It seems possible that a simpler task may have comparable results, and it would be an interesting direction for future research to investigate how each of these stages are contributing to learned object permanence. I'm not an expert in developmental psychology, but I know there is some research showing that smooth-motion visual signals are the main prerequisite for object permanence in chickens (https://onlinelibrary.wiley.com/doi/abs/10.1111/desc.12796). An ablation study over game mechanics or world properties could lead to some really interesting results.

Of course, the paper you've submitted already describes a huge volume of research work and doesn't need more experiments. Unfortunately, it is a bit difficult to follow as a result. You may want move some content from Fig. 3 and 4 (those figures are difficult to parse, even with my pdf zoomed in 250%) to the appendix, if only to focus the reader's attention on specific results. While arguing that your agent successfully plays the game is important, it should take a backseat to the experiments which probe the properties of your static and dynamic image representations.

---

> ### Author Response · Authors · 2020-11-17
> **Response to Reviewer #3**
>
> Thank you very much for your efforts in reviewing our work and for your support of our paper submission. We address your comments below.
>
> **Cache is complicated with many stages. A simpler task may have comparable results**
>
> While conceptually simple, Cache is indeed a non-trivial game to implement in an embodied environment, and has many stages. However this adversarial setup has the advantage of automatically generating diverse (and increasingly challenging tasks) goals for the agent to learn from. Without this, we would potentially need to hand define the curriculum for our agent. As noted in the paper, we will release our implementation for others to build upon, which will hopefully encourage other researchers to build more easily upon our work.
>
> Devising easier variants of Cache to explore which stages contribute what to the representations learnt is an exciting and possibly revealing area for future research. We hypothesize that each stage contributes preferentially to different downstream tasks (e.g. without object manipulation our agent would be unable to learn to track hidden objects while without the exploration/seeking stages the agent would likely not learn about free space). In future work we hope to understand the depths of interaction between these stages: e.g. does learning about free space provide a foundation upon which it is easier to learn about object permanence? In our comparisons in the Experimental section, we show that representations learned using Cache outperform representations learnt by a Navigation agent. While navigation is but a small and possibly less interesting aspect of cache, it is revealing to see that other aspects of cache are contributing significantly to the learned representations.
>
> **Figure rearrangement: (Same reply as given to Reviewer #1 since the concern was very similar)**
>
> This comment also reflects a concern made by another reviewer. The figures in our paper, particularly Fig 3 are very dense with information and the page limit imposed by the conference forced us to make them quite small. We address this concern by the following:
>
> 1. We provide better versions of these figures in the rebuttal PDF - this still restricts our sizes, but does provide an additional page to address the most affected parts of the figure
> 2. We will provide an accompanying webpage with high resolution figures which are easy to view and navigate at the time of creating a camera ready version of the paper, should it get accepted.

---

### Official Review · AnonReviewer4 · 2020-10-27
**Initial review**

**Rating:** 8
**Confidence:** 4

**Review:**

Summary
-------

This paper examines the representations learned during adversarial gameplay, specifically a hide-and-seek game called Cache.  The hiding agent must place an object in a room such that the seeker agent cannot find it.  The authors argue that the adversarial nature of the game shapes the representations.  Inspired by psychology experiments performed on children, the authors examine both static and dynamic representations to probe whether they contain information about properties of the environment such as object permanence.

Positives
---------

The paper addresses an interesting problem by studying representations learned by agents interacting in an environment.  These representations perform favorably compared to passive representations.

The paper is thorough, detailed, and well written.

The inclusion of dynamic representations is interesting, and the psychology-inspired experiments are adapted well to the Cache agent's environment.


Negatives
---------

My primary concern with the paper hinges on a trick the authors used to get their agent to learn meaningful representations.  The authors introduce Visual Dynamics Replay (VDR) to improve the agent's performance.  Essentially, this is a set of auxiliary tasks and self-supervision derived from the agent's interactions with the environment.  Although not a problem in itself, VDR was not used in the Navigation agent or Random agent used as baselines in the paper.  This leaves the conclusions of the paper much weaker.  It is unclear if these auxiliary tasks of VDR or the adversarial gameplay of the problem are the driving factor behind the learned representations.

Because the scope of the paper is so broad, it is difficult to evaluate some of the contributions.  For example, the perspective simulation module seems useful, but there are no ablation experiments evaluating changes to the model.


Reasons for Score
-----------------

Although the paper is intersting, detailed, and thorough, I have trouble with the strong conclusions drawn from the experiments because of the possible confounds of VDR.


Minor Issues that did not affect score
--------------------------------------

Graphs in figure 5 are difficult to see and interpret, even at full zoom on a monitor.


Update after rebuttal
------------------

I am very interested to see the results of the experiment mentioned in the rebuttal.  Although you mention that gameplay dynamics and ability to interact with objects is crucial, I still feel that this isn't as cleanly demonstrated as it could be without addressing these confounds.  However, in retrospect I also realize that my initial score suffered from tunnel vision on that single issue, which is important to several claims in the paper, but is by no means the only contribution.  The paper is ambitious in scope, novel, and well written, so I have increased my score.

---

> ### Author Response · Authors · 2020-11-17
> **Response to Reviewer #4**
>
> Thank you very much for your feedback and time spent reviewing our work. We address your comments below.
>
> **The importance of VDR**
>
> VDR is indeed along the same lines as a growing body of work in self supervised learning and learning with auxiliary tasks. However, note that VDR only directly affects SIRs. With regards to DIRs, it is possible that it may indirectly affect the ability to learn more powerful DIRs, but its effect is likely quite muted. Our comparison to the navigation agent was intended to be more of a comparison to show that high quality DIRs do not occur simply from popular existing tasks in embodied AI which focus on active navigation, but without the encouragement of gameplay dynamics and without the ability to interact with objects in one’s environment - both crucial and enabled by playing Cache. We hypothesize that adding VDR to the training of the navigation agent may improve its performance in the free space seriation experiments but as a navigation agent will never interact with objects it seems very unlikely to have any impact on the contained/behind or hidden object tracking experiments (note also that this hypothetical improvement in the free space seriation task would serve to underscore the value of VDR). Nevertheless, we will attempt such an experiment and report the results in a later revision. Unfortunately training such a navigation agent is quite computationally expensive and regenerating our analysis can only be done after this agent has trained: with machines at our disposal, these results will likely not be available before the end of the paper revision period.
>
> **More ablations**
>
> While we have provided an extensive experimental evaluation of our learned representations, as also pointed out by the other reviewers, we agree that further ablations would help highlight the impact of our design choices. The experiment mentioned above (adding VDR to a navigation agent) will help in this direction. Ablating each part of the model is possible, but extremely expensive owing to the long training times and consumed GPU resources. As a result, we will attempt to choose some critical and removable components of the model to perform an ablation analysis as part of future work. Another interesting direction for future work has been recommended by Reviewer 3 -- playing variants of the game of Cache to tease out how different parts of Cache lead to different representations. Both of these directions are natural choices for future research and we are excited about these directions.

---

### Official Review · AnonReviewer1 · 2020-10-27
**Review of Learning Flexible Visual Representations via Interactive Gameplay**

**Rating:** 8
**Confidence:** 4

**Review:**

In this paper, the author's propose an embodied adversarial reinforcement learning agent that can play a variation of hide-and-seek called Cache. This environment is a high fidelity interactive world. The authors argue that the agents are able to learn flexible representations of their observations which encode information such as object permanence, free space and containment.

The authors provide a well-written description of their game and provide well-designed visualizations to understand the interactions of the agent and the observations required for the learning problem. The authors present a concise and well-researched literature review which serves to distinguish their work as novel and well built on underlying prior work.

The authors make several contributions in this paper. First, an adversarial game Cache which permits the study of representation learning in the context of interactive visual gameplay. Furthermore, they present an agent which can perform strongly on the benchmark that they create even in comparison with human players. Finally, they present a study of the static and dynamic representations learned by the agent.

The authors provide an overview of the architecture for the Cache agent. It is well researched, well-reasoned and presented in a way that is easy to understand and reproduce.

The authors present multiple experiments in their paper and the corresponding deep Appendices. Specifically, they attempt to understand how agents can learn to proficiently hide and seek objects with static image representations and dynamic image representations

Of particular interest are the dynamic tasks the authors present in Section 5: Experiments. The authors make allusions to study of human children and object permanence. The results are quite compelling and well presented in a way that makes them easy to understand. I would urge the authors to consider if there are any non-ablative baselines against which they may be able to compare their model. One baseline that the author's use is human volunteer comparisons, these are compelling and presented in the appendix.

Figure 3 is perhaps the weakest figure in the paper. While it provides a great amount of information I would argue that the results are presented in a way that makes them less legible than if the figure was broken out into methods and results. The same could be said about Figure 4. The visualizations of the agents representations of synthetic and natural images are combined into a single image with performance on corresponding tasks. While I understand that the authors made these choices for space constraints, I would urge them to reconsider the visual hierarchy of the figures to emphasize the performance of their agents.

I feel like this is a paper that would be of great interest and benefit to the ICLR community.

---

> ### Author Response · Authors · 2020-11-17
> **Response to Reviewer #1**
>
> Thank you very much for your feedback and support of our paper submission. We address your comments below.
>
> **Non-ablative baselines**
>
> In Section 5, we provide comparisons to baselines in each of the three experimental analyses -- Evaluating the Cache agent at the game of Cache, evaluating Static Image Representations (SIRs), and evaluating Dynamic Image Representations (DIRs). Since the game of Cache has not been extensively studied in the past, significant baselines do not exist in the literature. As a result, the Cache playing ability of our agent was put into context in comparison with the ultimate baseline -- humans players. For the SIR and DIR evaluations however, we have provided comparisons to several other baselines:
>
> 1. an active embodied agent (Navigation),
> 2. a standard and well accepted task to train representations (image classification using the data collected by our Cache playing agent),
> 3. a standard self-supervised method (auto-encoder),
> 4. a randomly initialized agent, designed to primarily test if the proposed architecture has any inductive biases that are sufficient to produce powerful representations, and
> 5. the present day gold standard of representation learning (ImageNet) which has been provided primarily to put the above results in context, and can serve as a benchmark that we desire to attain.
>
> **Figure rearrangement**
>
> This comment also reflects a concern made by another reviewer. The figures in our paper, particularly Fig. 3 are very dense with information and the page limit imposed by the conference forced us to make them quite small. We address this concern by the following:
>
> 1. We provide better versions of these figures in the rebuttal PDF - this still restricts our sizes, but does provide an additional page to address the most affected parts of the figure.
> 2. We will provide an accompanying webpage with high resolution figures which are easy to view and navigate at the time of creating a camera ready version of the paper, should it get accepted.

---

> > ### Comment · AnonReviewer1 · 2020-11-24
> > **No change to rating, appreciate the comments and modifications for publication.**
> >
> > Thank you for the rebuttal and delineating these modifications.

---

### Official Review · AnonReviewer2 · 2020-10-29
**Convincing paper about learning transferrable reps from correlated/synthetic images via competitive interaction**

**Rating:** 9
**Confidence:** 3

**Review:**

The paper intends to contribute a novel task (Cache, as realized in AI2-THOR), the architecture of a strong Cache agent which learns reusable representations which allow significant transfer performance, and novel methods for evaluating the quality of dynamic image representations. The first and third contributions are directly related to the conference topics, and the second provides additional evidence in favor of the paper’s core idea: training on interactive gameplay allows learning flexible representations (in the sense of supporting many tasks via transfer) even when images are highly correlated and synthetic.

The key strength of the paper is the very general core idea it advances and how this idea is explored via a novel task. The paper is easy to read and convincing. The partition into details needed for the paper’s core argument and details specific to the experiments in the appendix is well done. (If anything even more could have been pushed to the appendix.)

One weakness of the paper is that a specific notion of “flexible” (which is mentioned in the title and twice in the abstract but nowhere else) is not advanced or integrated with the core idea. How does gameplay relate to flexibility? Why might flexibility be harder to achieve via passive learning or reinforcement learning with fixed reward functions? Because the authors place stress on the idea of how play and interaction contribute to representation learning (rather than a new method), slightly more space should be given to developing the general idea. The idea is not specific to vision, but only vision-related representations are considered. Sketching how the idea ought to work for text or audio would be useful if the focus really is on this very general idea.

Recommendation: strong accept. The philosophical aims of the paper make it stand out amongst the mass of related work that is otherwise very engineering focused. The experiments are soundly executed in a way that ends up clearly demonstrating the core idea.

Questions for the authors:
- A step where the hider needs to retrieve the object they hid would seem appropriate. Are there certain limitations of the AI2-THOR environment that make adding this step (which would seem to expose more of the richness of the simulated world through fixed rules of the game) infeasible to add?
- Inversely, do the authors feel that it was important that the hider manipulate the object into the desired location? How much of the richness of the simulated world comes through in the task feels relevant to the core ideas of the paper, but the paper currently does not address this kind of detail in the design of the Cache game within AI2-THOR.

Section-by-section reactions: (to see how opinions change over time)

Title+Abstract:
- The notion of “flexible” seems to be at the heart of this paper’s intended contribution. Hopefully it will be defined in the body text. Uh oh, it looks like “flex” only ever appears on the first of the submission’s 36 pages. Hopefully a synonym will get defined later.

Introduction:
- Excellent motivation.
- Good that representations of interest (SIRs/DIRs) are named and distinguished. Many other papers, in the interest of highlighting end-to-end training, would forget to do this.
- Good explicit list of contributions, excellent that two are specifically centered on representation learning.
- Missed opportunity to highlight a distinct role for “flexible” representations. (I don’t quite know what it should mean beyond supporting transfer well. A representation that could easily be scaled up or down in dimensionality by stripping channels in a well defined order might be considered flexible in another sense. Likewise, one that was defined in terms of pluggable input modules to work with novel combinations of familiar input types might be considered differently flexible. What kind of flexibility do you want?)

Related Work:
- Another take on learning visual representations via interactive gameplay is seen in https://arxiv.org/abs/1812.03125 where the authors learn a SIR (trained on a proxy task of predicting videogame memory state) that supports the use of low-continuous-space exploration strategies like rapidly-exploring random trees. The representations are learned offline/passively, but they are learned as to improve the efficiency of the very exploration process that builds that dataset for offline learning.

Playing Cache in a Simulation:
- It seems notable that the hiding agent is never asked to retrieve the object they have hidden. Without this step, the hiding agent may find ways of manipulating objects in a way that makes them simply unretrievable (e.g. the object is pushed into a corner in a way that causes it to glitch out of the room, etc.). A step like this would require the hider to learn a finer grained representation of the hiding location that gives itself a clue as to how it should be retrieved (e.g. “under the couch in a place you’ll never be able to see but will be there if you actually reach for it).

Learning to Play Cache:
- Great!

Experiments:
- All well done.

Discussion:
- “We believe that it is time for a paradigm shift via a move towards experiential,
interactive, learning.” -- something similar has been said by many other researchers in many different decades, so it would be good to say what’s different about the situation in 2021. The difference now seems to be the availability of simulators with visual fidelity comparable enough to reality to demonstrate meaningful sim2real transfer. Are there other bullet points that could be added to a why-now argument?

---

> ### Author Response · Authors · 2020-11-17
> **Response to Reviewer #2**
>
> Thank you very much for your feedback and support of our paper submission. We address your comments below.
>
> **Definition of flexible**
>
> We have used the word flexible to refer to the generalizabilty and utility of these learned representation stacks in downstream tasks, such as our developmental psychology motivated experiments. The word flexible has been used since the representations were not trained using tasks similar to the downstream applications. For example, this work does not train representations using classification and then show generalizability to other classification tasks, whether in domain or out of domain. Having said that, after reading your comments, we do agree that flexible is likely a strong expression to attribute to these representations, particularly in the absence of a formal definition, and hence we will be rewording this to “generalizable”.
>
> **Developing the general idea**
>
> In this work, we have focussed on learning visual representations via gameplay, but we do agree with you, that gameplay may be incredibly powerful to learn representations required for learning text and audio representations. However, since the authors on this paper do not consider themselves as domain experts in either of those fields, we have avoided making bold and general claims in these directions. We have, however, now added a few lines to our discussion section.
>
> **A step where the hider needs to retrieve the object**
>
> We considered this in our original design. Basically the idea was to score the hider as ("seeker performance with knowledge of the object's location" MINUS "seeker performance without this knowledge"). In practice it is true that the hider can place the object in locations that the seeker cannot, in principle, find but this happened infrequently enough during training (and even with human participants!) that we did not implement it. As we develop more nuanced models capable of playing cache, this will likely help quite a bit to learn a finer grained representation of the hiding location as you have mentioned.
>
> **Is object manipulation important**
>
> We do feel that this is important. Without being able to directly manipulate/move objects it seems unlikely that the agent would ever be able to develop an understanding of objects' physical properties and affordances. Indeed, in future iterations of this task we hope to allow and encourage even more direct interaction of multiple objects, e.g. allowing an agent to hide an object by placing another object on top of it or in front of it. This, we hope, will enable the agent to learn representations that are able to encode relative sizes, volumes, etc. in ways more powerful than the current method enables.
>
> **Suggested reference:**
>
> Thank you for this recommendation. This paper is indeed relevant. We have added this to our Related Work section.
>
> **Why-now**
>
> We believe that it is time for a paradigm shift via a move towards experiential, interactive, learning for at least two reasons:
>
> 1. After huge progress in supervised learning, the AI community has recently made huge progress towards self supervised learning, in both computer vision and natural language processing. This is a good time to build upon this growing interest and explore the direction of experiential learning.
> 2. Exploring this direction at the moment is infeasible with real robots, owing to the high costs of acquiring and maintaining robots, data inefficient learning algorithms, as well as the slow nature of physical robots. However, recently several simulators have been developed (including AI2-THOR which we have used here, Habitat, Gibson, Sapien, etc.).  These simulators are getting more photorealistic and continue to add more capabilities such as physics realism. The presence of these fast and cheap simulators makes it an opportune time to explore this direction.
>
> We have added this to the discussion section.

---

### Decision · Program_Chairs · 2021-01-07
**Final Decision**

**Decision:**

Accept (Oral)

**Comment:**

Motivated by the importance of gameplay in the development of critical skills for humans and other biological species, this work aims to explore representation learning via gameplay in a realistic, high fidelity environment. Inspired by childhood psychology, they propose a variant of hide-and-seek game called "Cache" built on top of AI2-THOR, where one agent must place an object in a room such that another agent cannot find it, and demonstrate that the adversarial nature of the game helps the agents learn useful representations of the environment. They examine the difference in representations learned via such a dynamic, interactive adversarial gameplay approach, vs other more passive approaches involving static images.

The paper is well written and motivated, and easy to follow. All reviewers agree that the paper will be a great contribution to the ICLR community. I believe this is an important work, because not only does it challenge the traditional way of training many components of our systems passively (via static image recognition models), it synthesizes ideas from various disciplines (psychology, embodiment, ML) and provides an excellent framework for future research. For these reasons I'm recommending we accept this work as an Oral presentation.